Registered report

psychology/ecology/behaviour

personality, social thermoregulation, attachment, machine learning

**Author for correspondence:**
Hans IJzerman
e-mail: h.ijzerman@gmail.com

# Individual differences in adapting to temperature in French students are only related to attachment avoidance and loneliness

Adrien Wittmann[1], Mae Braud[1], Olivier Dujols[1], Patrick Forscher[1,2] and Hans IJzerman[1,3,†]

[1]Laboratoire InterUniversitaire de Psychologie. Personnalité, Cognition, Changement Social, Université Grenoble Alpes, Saint-Martin-d'Heres, Rhône-Alpes, France
[2]Busara Center for Behavioral Economics, Kenya
[3]Institut Universitaire de France, Paris, France

HI, 0000-0002-0990-2276

Among animals, natural selection has resulted in a broad array of behavioural strategies to maintain core body temperature in a relatively narrow range. One important temperature regulation strategy is *social thermoregulation*, which is often done by warming the body together with conspecifics. The literature suggests that the same selection pressures that apply to other animals also apply to humans, producing individual differences in the tendency to socially thermoregulate. We wanted to investigate whether differences in social thermoregulation desires extend to other personality factors in a sample of French students. We conducted an exploratory, hypothesis-generating *cross-sectional* project to examine associations between thermoregulation and personality. We used conditional random forests in a training segment of our dataset to identify clusters of variables most likely to be shaped by individual differences to thermoregulate. We used the resulting clusters to fit hypothesis-generating mediation models. After we replicated the relationships in two datasets, personality was not related to social thermoregulation desires, with the exception of attachment avoidance. Attachment avoidance in turn predicted loneliness. This mediation proved robust across all three datasets. As our cross-sectional studies allow limited causal inferences, we suggest investing into prospective studies to understand whether and how social thermoregulation shapes attachment avoidance early in life and loneliness later in life. We also recommend replication of the current relationships in other climates, countries, and age groups.

†Present address: Laboratoire InterUniversitaire de Psychologie, BP47 – 38040, Grenoble – Cedex 9, France.

# 1. Individual differences in adapting to temperature in French students are only related to attachment avoidance and loneliness

In the animal kingdom, successfully managing core body temperature strongly influences the probability of survival. For this reason, animal species have evolved a broad array of evolutionary strategies for managing their core body temperature—including behavioural strategies. Prime among these behavioural strategies, and a strategy on which human and non-human animals alike rely, is leveraging the body heat of conspecifics through huddling and hugging [1–3]. This *social thermoregulation* has been particularly important in humans to adapt to the cold throughout evolution. Yet some succeed in coping with the environment better than others. As a logical consequence, individual differences should exist in adapting to climate through the individuals' preferences to manage cold socially.

We conducted an exploratory, hypothesis-generating study to investigate whether preferences for social strategies to cope with temperature—that probably differ between individuals—relate to how people differ in their personalities. The idea behind this work is that the influence of social thermoregulation on personality would be transmitted by people's past histories with close relationships (their *attachment style*; [4]), because this past history will determine whether people feel that relying on others for temperature regulation is a safe strategy.

Although they are grounded in sensible theories of evolution and relationships, our ideas are admittedly speculative. Thus, we relied on an exploratory analysis strategy enabled by a machine learning technique known as conditional random forests to generate plausible hypothesis-generating mediation models from a training dataset. We then test the replicability of these models using a testing dataset and a second replication dataset after review with this journal.

## 1.1. Social thermoregulation to adapt to the environment

Across the animal kingdom, relying on others to regulate core body temperature is often privileged over internal methods, such as shivering [5]. This is because methods such as shivering, while effective at producing heat, rely on the body's internal (and expensive) metabolic processes, whereas social methods of temperature regulation do not. Emperor penguins, for example, huddle for warmth, reducing the surface area exposed to cold and allowing colder penguins to rely on the heat from warmer conspecifics. The result is that thermoregulation via huddling requires approximately 38% less energy to warm their bodies than when penguins do it when alone [2].

This same evolutionary logic, that social methods of regulating core body temperature are much more efficient than internal processes, applies just as strongly to humans as it does to penguins [6]. Human behaviour should therefore bear the imprint of adaptations to maintain core body temperature through the behavioural use of conspecifics. Even though modern technology has made the regulation of ambient temperature an everyday convenience for many people, these technologies have existed for too short a timeframe to eliminate these behavioural adaptations.

Some evidence does indeed suggest that humans use others to help regulate their core body temperature. For example, common brain regions are involved in the regulation of social interactions and core body temperature [7,8]. The concepts of temperature and sociality also appear cognitively linked: social proximity and distance influence our perceptions related to temperature [9,10] and people tend to think more about their loved ones when they are physically cold [11].[1] In the latter project, people with a secure outlook on relationships were more likely to think of loved ones when cold, whereas those with an avoidant outlook were less likely.

## 1.2. Individual differences in dealing with the environment across the animal kingdom

Even in the animal kingdom, individual animals of a particular species differ in their strategies for coping with the environment, including regulating their temperature. In animals, these individual differences

[1]Notably, we recognize that the psychological sciences are facing a replication crisis. This means that—like most of the psychological literature—the social thermoregulation literature has been confronted with conflicting findings: some effects were successfully replicated [12,13], whereas others were not [14,15]. Some effects relied on sufficiently large samples [11,16–18], while others were tested using underpowered designs [19–21]. An in-progress meta-analysis that corrects as well as possible for bias in the literature suggests, however, that the effects in this literature are probably not due to selective reporting alone (H IJzerman, R Hadi, N Coles, E Sarda, R Klein, I Ropovik 2020, unpublished manuscript).

are known as life-history strategies [22,23]; such strategies generally manifest as stable differences in resource gathering, reproduction, and the avoidance of predators [22]. Life-history strategies generally fall into two broad clusters, or temperaments, one of which involves proactive strategies and one of which involves reactive strategies. Proactive animals engage in more aggressive, competitive and risk-taking behaviours, whereas reactive animals engage in more passive, cooperative, risk-averse behaviours. The proactive trait usually emerges in more stable environments, whereas the reactive trait tends to be favoured in unpredictable environments [24,25].

Clusters of individual differences similar to animal temperaments also manifest in humans. According to predictive and reactive control systems (PARCS) theory, people differ in their tendency to rely on predictive versus reactive control—two systems that resemble proactive and reactive animal temperaments depending on the predictability of the environment [25,26]. An environment is thought to be predictable when it is not threatening or when threats are manageable (possibly including, for example, bodily harm, harsh climates or limited economic means). People engage more in predictive control when processing familiar stimuli and tend to engage more if they are raised in predictable environments. People engage more in reactive control when processing novel information and if they are raised in more unpredictable environments. Predictive control is considered more metabolically efficient than reactive control because it allows people to schedule more in advance [25].

The strong resemblance between the major clusters of non-human individual differences and the major clusters of human personality suggests that stable individual differences in human and non-human animals may be clustered to solve similar adaptive problems. Thus, just as animal temperaments are shaped by and adapted for particular environments, human personality may be shaped by and adapted for particular environments. This suggests that similar forces may shape human and non-human personality. It follows that strategies increasing the environment's predictability may come into play in personality processes.

## 1.3. Individual differences in coping socially with environmental demands

Understanding how the predictability of the environment affects human personality processes requires identifying the parameters of predictability. The first of these is how people can cope with the environment. A primary feature of the environment's predictability is the availability of reliable conspecifics. Bowlby [27] postulated in his *Attachment Theory* that early relationships are essential for managing the environment, as early relationships promote survival. Carers fulfil a number of functions toward the child, notably protection against environmental threats, nurturing, and warming up when cold. These first experiences turn into individual differences predicting others' behaviours, and, according to how people think that others are reliable, into more general personality differences (e.g. [28]).

A second parameter of predictability concerns *which* threat the environment poses. Temperature is only second in importance next to oxygen regulation for survival. Yet little attention has been devoted to directly measuring how humans deal with temperature, and even less so how they deal socially with the environment, with the exception of one tool giving a first impulse in this direction: the *Social Thermoregulation and Risk Avoidance Questionnaire* (STRAQ-1; [29]). The STRAQ-1 has for example a subscale that reliably assesses individual differences in terms of desires to socially thermoregulate. The STRAQ-1 was created and validated to investigate whether people's strategies to thermoregulate relate to their feelings of reliability and safety in relationships. And indeed, Vergara *et al*. [29] found that people's desires to socially and solitarily thermoregulate, respectively, relate to avoidance and anxiety in relationships. The STRAQ-1 thus provides not only a valid measurement tool, but also shows that people's ways of coping with the environment relate to their attachment styles. This is thus a first indication that the way people cope with the environment could potentially shape their personality.

## 1.4. How personality could help adapt to the environment

More commonly speaking, personality refers to an enduring pattern of thoughts, feelings and behaviours displayed by an individual across a variety of situations, which is supposed to be stable over time [30]. The most robust and famous personality tool used to understand personality is the *Big Five* factor model. The Big Five is a dimensional approach that assumes that personality may be understood in terms of five dimensions, where individuals score relatively high or low on a continuum on each dimension [31]. The five dimensions consist of Openness (referring to the propensity of an individual to be open to new experiences and creative), Conscientiousness (referring to discipline and drive towards achievement),

Extraversion (referring to assertiveness and gregariousness), Agreeableness (referring to kindness and compliance), and Neuroticism (referring to individuals' tendency to feel depressed or anxious; [31]).

Only in the last 20 years, a consensus emerged that personality in humans also may have an adaptive function and that it may be understood as life-history behavioural strategies [22,23,32]. In line with this postulate, personality has been shown to be influenced by the 'socioecological' environment to ensure better pay-offs [33], raising the question of whether temperature could have similar influences. Only in the last few years, a group of researchers found temperatures to be associated with personality. More specifically, Wei *et al.* [34] found that people are more agreeable, extraverted, conscientious, open to experience, and less neurotic if they grew up in clement climates (closer to the psychophysiological comfort optimum of 22°C). This tentatively suggests that climate influences personality. To the extent that personality is a type of life-history behavioural strategy, personality should also be influenced by the strategies that people adopt to cope with fluctuations in local temperature.

The Big Five captures only one aspect of personality. If individual differences in temperature-coping strategies are related to personality at a broad level, these strategies may be linked to personality aspects not captured by the Big Five. Some have argued, for example, that the five-factor personality model can be captured in two higher-order factors (stability and plasticity; [35]). Other researchers have favoured a six-factor model (the HEXACO; [36]). In forager-farmers in the Bolivian Amazon, only three instead of five factors emerged [37]. In yet another (dictionary-based) study in Farsi, a Big Two or Big Three structure emerged [38]. In a dictionary study of the French personality lexicon, six factors emerged (albeit not the same as the HEXACO; [39]). Only one 2015 study in a French sample did detect a five-factor structure of a personality inventory for the DSM [40]. Altogether this meant that some uncertainty exists as to which model is the most likely to emerge from our sample of French students.

As well, we remain quite uncertain how temperature may have shaped human life-history strategies for the mere reason that, unlike ecologists, psychologists have not paid as much attention to the influence of temperature. Also, we do not know exactly how to measure 'life-history strategies' in humans, thus supplying the need to step back and explore when testing our general ideas in a French sample. Taking all these arguments into account, prior to moving on to our main exploratory–confirmatory analysis in which desires to (socially) thermoregulate are related to attachment and personality, we took an explicit, exploratory approach with few *a priori* assumptions. This meant that we first selected a large number of items in our training dataset and conducted an exploratory factor analysis to create new personality variables [41,42]. At the end of our analyses section, we do, however, test similar mediation models with the traditional factor structures of the assessed scales.

## 2. Research overview

Overall, we suspected that individual differences in personality variables could be related to individual differences in the way people deal with temperature constraints. We also suspected that people are more likely to use social strategies to regulate their temperature in predictable social environments. We assumed that attachment mediates the relationship between the regulation of temperature and personality.

In our work, we only provide starting points for a process of guided exploration by conducting hypothesis-generating research in two cross-sectional datasets. We used conditional random forests, a machine learning technique specialized for exploratory research, to identify robust relationships in our dataset and fit a series of mediation models.

These mediation models cannot be taken as evidence of causal mechanisms given that our data are cross-sectional and that mediational processes we assumed imply changes over time. For example, Maxwell & Cole [43] demonstrated mathematically that the conditions under which cross-sectional designs may capture the true parameters of the mediation are rarely (if not never) met. Yet, despite the fact that our meditation does not reflect the causal processes we predict, they can indicate that some relations exist, particularly if replicated with the precision we predicted. Then still, a finding of statistical mediation in cross-sectional data does not imply the presence of causal relationships unless the analyst is willing to make strong, and unlikely unsupportable [43,44], causal assumptions. This project thus aims to identify mediated relationships that *could* exist rather than relationships that *do* exist. The relationships that we identify can then be investigated in longitudinal and experimental studies that support stronger causal inference, in which some predictions may pan out and others will fail.

Altogether, this study aimed to explore the potential mediation pathways through which individual differences to thermoregulate might impact personality through individual differences in attachment.

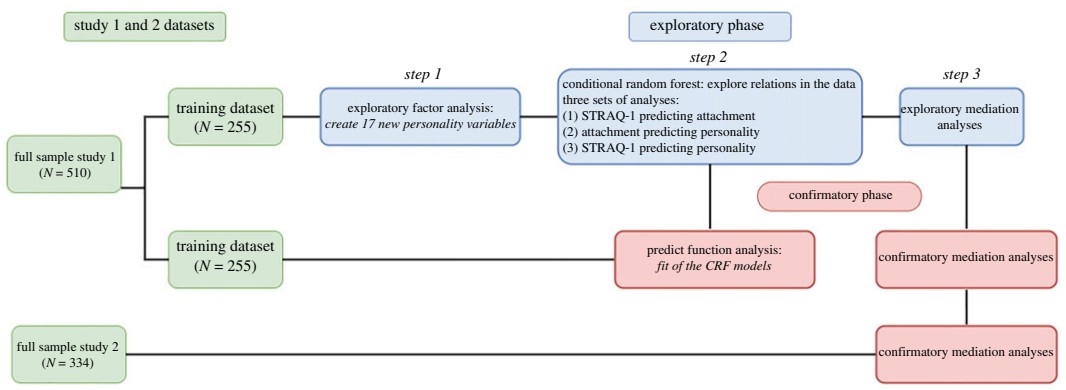

**Figure 1.** Analysis strategy of our study including the initial split of the dataset, the exploratory phase, and then the confirmatory phase.

In all our models, social thermoregulation (as measured by the STRAQ-1) served as the ultimate cause, attachment (as measured by Experiences in Close Relationships (ECR-R)) served as the mediator, and some aspect of personality served as the dependent variable personality. The analysis strategy follows three phases (figure 1): (1) Exploratory phase, in the training dataset of Study 1 we used conditional random forests to select relevant variables for the mediation models, and we tested the models; (2) Prediction phase, based on the parameters of the mediation models we then formalized predictions; (3) Confirmatory phase, we tested the replicability of our models in the testing dataset of Study 1, and in the full dataset of Study 2. As per the guidelines for Registered Reports, we had not yet analysed the confirmatory set of Study 1 prior to pre-registering them and having them reviewed. These analyses and the replication results of the full dataset of Study 2 and our discussion were added after review. The analysis workflow is graphically depicted in figure 1.

# 3. Study 1

This manuscript was generated via papaja [45] and was conducted in line with the CO-RE Lab Lab Philosophy v. 4 [46].

## 3.1. Participants and procedure

For our first study, we relied on a convenience sample of already collected data. For this sample, participants ($N = 510$) were recruited from Université Grenoble Alpes in France in 2018 and answered multiple online questionnaires on Qualtrics. Participants were psychology students (69 men, 435 women, and 6 others): 134 in L1, 192 in L2, 132 in L3 and 50 in M1 (3 missing). The $M_{\text{age}}$ was 20.89 ($s.d._{\text{age}} = 6.08$), $M_{\text{self-reported height}}$ was 166.35 cm ($s.d._{\text{self-reported height}} = 8.14$ cm)[2] and the $M_{\text{self-reported weight}}$ was 59.71 kg ($s.d._{\text{self-reported weight}} = 11.46$ kg). Amongst our participants 123 were smokers and 382 were not (4 missing). Also, 281 were in a relationship, 213 were not, and 16 preferred not to answer.

## 3.2. Power analysis

To ensure we had sufficient power to run mediations in either half of our sample, before we conducted our analyses, we ran an *a priori* power analysis for mediation based on the Sobel test with the *powerMediation* package [47]. It first required us to estimate the standard deviation of the independent variable and of the mediator. It also required us to estimate the effect size of the relations between the independent variable and the relations between the mediator and the dependent variable. Finally, it required us to compute the standard deviation of the error term of the relation between the independent variable and the dependent variable controlling for the mediator based on the effect size of the three paths of the mediation (for more information, see [48]). Based on a minimum effect size of interest of *beta* = 0.25 for

---

[2]We removed observations under 100 cm for the calculation of the $M_{\text{self-reported height}}$ and $s.d._{\text{self-reported height}}$, given the high unlikelihood of these scores (and were probably in our data due to typing errors).

all relations in the mediation and a statistical power of 0.80, we found that 231 participants were needed to detect a mediation. We thus concluded that our number of participants was sufficient to be able to split our data in two equal parts (for our script and other details, see https://osf.io/74fr3/).

## 3.3. Measures

Our measures were chosen by researchers who participated in the so-called 'testweek' at Université Grenoble Alpes. Department members nominated questionnaires for inclusion (as these were thus not chosen by us, it constrained what we could find in our sample). The department then posted Qualtrics links with the full battery of questionnaires to student Facebook groups. Students could then participate in a battery of questions in exchange for course credits. Before moving on to our exploratory factor analyses, we report the scales in their usual form together with their reliability from the training set (table 1; these were updated to include the test set after review with this journal).

## 3.4. Data preparation: exploratory factor analysis

### 3.4.1. Personality variables

Because of the relative uncertainty of the factor structure for our sample, we formed new aggregate measures using exploratory factor analysis. This method allowed to create scales that are more valid in terms of construct validity and provide explanations that are more economical [41,42]. Admittedly, this is based on a convenience sampling, as we did not select these items *a priori*. But by grouping items sharing the highest amount of variance between scales, the constructs measured are more likely to measure a single trait that does not overlap with other constructs.

From our dataset, we inserted items from the following variables into our exploratory factor analysis: Attachment Anxiety, Attachment Avoidance [59], Repetitive Thinking Mode [55], the Big Five factors Openness, Agreeableness, Conscientiousness, Extraversion and Neuroticism [49], Speciesism [50], Self-esteem [53], Self-control [54], Self-Reported Stress [58], Prejudice Toward North Africans [51], the UCLA Loneliness scale [60], Right Wing Authoritarianism [61], Social Dominance Orientation [52], DASS Depression, Anxiety, and Stress [56], and Well-being [57].

In order to determine the number of factors from our items, we used both parallel analysis [62,63] and Velicer's [64] MAP test. Our parallel analysis suggested a 17-factor solution, whereas the Velicer's MAP provided similar average squared partial correlation values for 17–19 factors and higher for 16 factors. Based on the results of both tests (and choosing the least amount of factors for reasons of parsimony), we extracted 17 factors for our further analyses.

To create each of the 17 scales, we first examined factor loading of each individual item as we only included items if the factor load was superior to 0.30. In addition, we inspected each item to understand how applicable the item was to the construct (for example, we left out 'I love life' for the self-esteem measure). Finally, we removed any items that cross-loaded onto multiple factors in order to reduce overlap between the created scales [65]. All the excluded items with rationale for exclusion are presented in appendix A. With the final list, we averaged items to 17 different scale averages.

Our 17 new scales, we titled Attachment Anxiety 'Modified' (which dropped two items of the original scale), Attachment Avoidance 'Modified' (exact same scale as the original), Anxiousness, Well-being, Self-esteem, Self-discipline, Loneliness 'Modified', Social Dominance Orientation 'Modified', Right-Wing Prejudice, Impulsiveness, Reflection, Trust, Right-Wing Authoritarianism 'Modified', Stimulation, Sociability, Leadership, and Empathy. All new scales with their reliabilities and their composing items with their factor loadings are provided in appendix B. All the details of our factor analysis with all items and loads onto the 17 factors can be found on our OSF page: https://osf.io/f6qun/.

## 3.5. Analysis strategy overview

Our analytic strategy consisted of an exploratory phase and a confirmatory phase. Prior to analysing our data, we randomly split our dataset into a training and testing dataset. In the first phase of the exploratory analyses, we ran an exploratory factor analysis with all the variables in our dataset, except

**Table 1.** Scales in their usual form with their reliabilities from the training set. *Note.* Alpha is Cronbach's alpha. Omega is McDonald's omega total.

| scales | subscales | items | alpha | omega | M | s.d. | example item | minimum | maximum | references |
|---|---|---|---|---|---|---|---|---|---|---|
| IPIP-NEO | openness | 24 | 0.79 | 0.83 | 3.65 | 1.12 | I prefer variety to routine | 1 (Very inaccurate) | 5 (Very accurate) | Johnson [49] |
| | conscientiousness | 24 | 0.88 | 0.90 | 3.46 | 1.10 | I carry out my plans | 1 (Very inaccurate) | 5 (Very accurate) | |
| | extraversion | 24 | 0.84 | 0.88 | 3.23 | 1.17 | I feel comfortable around people | 1 (Very inaccurate) | 5 (Very accurate) | |
| | agreeableness | 24 | 0.83 | 0.87 | 3.93 | 1.00 | I am concerned about others | 1 (Very inaccurate) | 5 (Very accurate) | |
| | neuroticism | 24 | 0.88 | 0.90 | 3.07 | 1.22 | I worry about things | 1 (Very inaccurate) | 5 (Very accurate) | |
| The Speciesism scale | | 6 | 0.73 | 0.77 | 2.10 | 1.32 | Morally, animals always count for less than humans | 1 (Strongly disagree) | 7 (Strongly agree) | Caviola et al. [50] |
| Prejudice toward North Africans | | 15 | 0.91 | 0.93 | 3.06 | 1.54 | Northern Africans have a culture too different from that of the French to be perfectly integrated in France | 1 (No, strongly disagree) | 7 (Yes, strongly agree) | Dambrun & Guimond [51] |
| The Social Dominance Orientation | | 16 | 0.89 | 0.92 | 5.71 | 1.33 | The lower groups should stay in their place | 1 (Strongly disagree) | 7 (Strongly agree) | Duarte et al. [52] |
| The Right-Wing Authoritarianism | | 20 | 0.87 | 0.90 | 2.50 | 1.43 | Our country needs a powerful leader, in order to destroy the radical and immoral currents prevailing in society today | −4 (Strongly Disagree) | 4 (Strongly agree) | Altemeyer et al. [39] |
| The Rosenberg Self-Esteem | | 10 | 0.90 | 0.92 | 2.22 | 0.79 | On the whole, I am satisfied with myself | 1 (Strongly agree) | 5 (Strongly disagree) | Robins et al. [53] |

*(Continued.)*

**Table 1.** (*Continued.*)

| scales | subscales | items | alpha | omega | M | s.d. | example item | minimum | maximum | references |
|---|---|---|---|---|---|---|---|---|---|---|
| The Self-Control Scale | | 13 | 0.81 | 0.85 | 2.86 | 1.13 | I am good at resisting temptation | 1 (Not at all) | 5 (Very much) | Tangney et al. [54] |
| The Repetitive Thinking Mode Questionnaire | | 18 | 0.62 | 0.77 | 2.62 | 0.86 | Once I started thinking about the situation, I couldn't stop | 1 (Not true at all) | to 5 (Very true) | McEvoy et al. [55] |
| The Depression Anxiety Stress Scale | | 21 | 0.93 | 0.94 | 2.01 | 0.94 | I couldn't seem to experience any positive feeling at all | 0 (Did not apply to me at all) | 3 (Applied to me very much, or most of the time) | Lovibond & Lovibond [56] |
| The Warwick-Edinburgh Mental Well-Being Scale | | 14 | 0.89 | 0.92 | 3.43 | 0.89 | I've been feeling optimistic about the future | 1 (none of the time) | 5 (all of the time) | Tennant et al. [57] |
| The Self-Reported Stress questionnaire | | 14 | 0.89 | 0.91 | 2.90 | 1.03 | In the last month, how often have you felt nervous and 'stressed'? | 1 (Never) | 4 (Very often) | Cohen et al. [58] |
| The Experiences in Close Relationships (ECR) | anxiety | 18 | 0.90 | 0.91 | 3.75 | 1.77 | I'm afraid that this person may abandon me | 1 (Strongly agree) | 7 (Strongly disagree) | Fraley et al. [59] |
| | avoidance | 18 | 0.94 | 0.95 | 2.83 | 1.65 | I usually discuss my problems and concerns with this person | 1 (Strongly agree) | 7 (Strongly disagree) | |
| The Social Thermoregulation and Risk Avoidance Questionnaire (STRAQ-1) | high temperature sensitivity | 7 | 0.85 | 0.9 | 3.09 | 1.24 | I am sensitive to heat | 1 (Strongly agree) | 5 (Strongly disagree) | Vergara et al. [29] |
| | social thermoregulation | 5 | 0.79 | 0.83 | 3.05 | 1.25 | When I feel cold I seek someone to cuddle with | 1 (Strongly agree) | 5 (Strongly disagree) | |
| | solitary thermoregulation | 8 | 0.73 | 0.81 | 3.37 | 1.20 | When it is cold, I wear more clothing than others | 1 (Strongly agree) | 5 (Strongly disagree) | |
| | risk avoidance | 3 | 0.54 | 0.59 | 3.52 | 1.14 | I try to maintain myself in familiar places | 1 (Strongly agree) | 5 (Strongly disagree) | |
| The UCLA Loneliness Scale | | 20 | 0.92 | 0.93 | 3.06 | 0.89 | I feel isolated from others | 1 (Never) | 4 (Often) | Russell [60] |

for the STRAQ-1. Based on factor loading, we created new personality variables, thus composed of items that loaded onto the same factors.[3]

Once we defined our new factors, we explored existing relations in our data to generate mediation model hypotheses through a powerful supervised machine learning method called conditional random forests. In supervised machine learning more generally, the algorithm infers a pattern from the data derived from a 'signal' (or dependent variable). The method relies on 'out-of-bag estimates', which involve repeated sampling from a training dataset (e.g. [67]). Multiple 'trees' are formed by assessing whether each variable influences the 'signal'. The 'trees' (votes on whether variables matter for the outcome variable or not) are then assembled into a 'forest'. Each 'tree' receives a 'vote' into an ensembled model that then summarizes all information from the trees. The outcome in the case of conditional random forests is a variable importance list. The importance list allows us to identify which are the best predictors of the variable of interest and which of the computed variables differ from random noise when predicting the variable of interest (see also [68]). In our case, the random forest allows us to select the variables to be included in the mediation analyses in the exploratory phase. In the first stage of our exploratory analyses, we ran three sets of conditional random forests, one for each of the three mediation paths (the $a$, $b$, and $c$ path).

In each set of analyses, we specified an outcome variable and derived the best predictor for the given outcome variable. To identify the mediators between thermoregulation (independent variables) and personality (dependent variables), we ran conditional random forests on the attachment variables (mediators) with thermoregulation variables as predictors to examine relations between the mediators and the independent variables ($a$ path in the model). Then, we ran conditional random forests on personality variables with attachment variables to select only personality variables that were related to attachment ($b$ path in model). After that we refined our personality variables selection by running conditional random forests on personality variables with thermoregulation variables as predictors to determine which personality variables were related to thermoregulation variables ($c$ path in the model) and thus to select only personality variables that were related to both attachment and thermoregulation variables. This funnel-like strategy enabled us to select the most relevant personality variable for the mediation analyses that followed and thus to generate the strongest mediation models to test, in a data-driven way. In the second stage of the exploratory phase, we tested the significance of these mediation models through more common, confirmatory, mediation analyses.

## 3.6. Main analysis plan: approach for the conditional random forests

All our analysis scripts, auxiliary results, and data can be found on our OSF project page: https://osf.io/74fr3/. In the exploratory phase, we first explored relations between the thermoregulation scales of the STRAQ-1 (social thermoregulation, temperature sensitivity, and solitary thermoregulation), attachment, and various personality factors via conditional random forests along the line of a mediation logic. As there were a large number of potential mediation models we could run (i.e. 2 independent variables × 2 mediator × 17 dependent variables = 68 potential models), conditional random forests helped us to determine which ones would be potentially relevant based on the three paths of the mediation (figure 2).

We chose conditional random forests over a regression for three reasons. First, linear regression is a parametric approach that requires *a priori* predictions of relationships between variables and also requires hypotheses regarding potential nonlinearities and interactions [69]. As we were interested in exploring our data, we relied on a non-parametric type of machine learning, which relies on a flexible number of parameters, where the number of parameters can grow as the algorithm learns more data [70]. Overall, they are therefore well suited for situations in which researchers have no *a priori* predictions such as exploratory research. Second, random forests are less prone to overfitting in relatively small samples with multiple variables [69]. Third and finally, random forests have a smaller chance for collinearity when including multiple predictors, as is the case in the current situation [71].

In order to understand which mediation models were most likely to be accurate representations of reality, we went through three sets of conditional random forests following the three steps of the

---

[3]Notably, we are conscious of the discussion of the contribution of states and traits to psychological measurements. For the sake of simplicity at this initial stage, we included them as equivalent in the factor analyses (without any decomposition into traits or states; [66]).

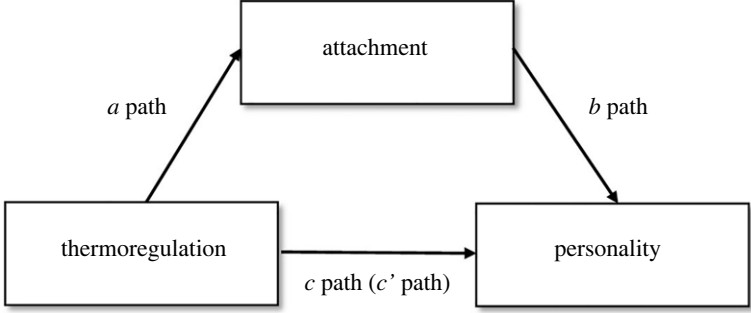

**Figure 2.** Mediation model and paths for thermoregulation, attachment, and personality.

expected mediation: (1) thermoregulation-related variables predicting attachment variables, (2) attachment variables predicting personality variables, (3) the thermoregulation variables predicting the personality variables. In each set of the conditional random forests, we added all the created personality variables as predictors of 'non-interest' for the main reason that the random noise is better estimated with an increased number of variables.

# 4. Exploratory results

## 4.1. STRAQ-1 predicting attachment (mediation path *a*)

To understand what predicted attachment, we ran one set of 100 conditional random forests for the newly created Attachment Anxiety 'Modified' as 'signal' and one set of 100 conditional random forests for the newly created Attachment Avoidance 'Modified' as 'signal' with the STRAQ-1 variables (excluding the Risk Avoidance scale)[4] as predictors. In order to be able to separate relevant predictors from noise, we added other personality variables that we were not interested in as variables of non-interest. In both of these sets of analyses, we also added in the other attachment variable that was not of main interest to be predicted (i.e. Attachment Avoidance 'Modified' when predicting Anxiety 'Modified' and vice versa) as they usually share considerable variance. By adding them, we could capture their unique variance predicted by the STRAQ-1.

   Here, we only summarize the predictors that differed from random noise. Results of the conditional random forests predicting the two newly formed attachment variables are displayed in the first section of table 2. In this table, we only report the extracted variables. The full variable importance can be found on our OSF page (https://osf.io/fchyd/). From the average of our first 100 conditional random forests predicting Attachment Anxiety 'Modified', we found that the eighth most potent predictor (after, amongst others, Loneliness 'Modified', Attachment Avoidance 'Modified', and Impulsiveness) of Attachment Anxiety 'Modified' was Solitary Thermoregulation. The most potent predictor from our second set of 100 conditional random forests predicting Attachment Avoidance 'Modified' was social thermoregulation (before, among others, Loneliness 'Modified', Attachment Anxiety, and Self-Esteem). Furthermore, Solitary Thermoregulation did not predict Attachment Avoidance and Social Thermoregulation did not predicted Attachment Anxiety. Thus, when using Attachment Anxiety as mediator, Solitary (and not Social) Thermoregulation should be used as predictor. When Attachment Avoidance is used as a mediator, Social (and not Solitary) Thermoregulation should be used as predictor.

## 4.2. Attachment predicting personality (mediation path *b*)

For the dependent variables/signals (the different personality factors we identified in our exploratory factor analysis), we ran 100 conditional random forests per variable with the newly created attachment variables. We again inserted the personality variables we were not predicting as variables of non-interest, but excluded the STRAQ-1 variables (as they were potential predictors of attachment in our mediation models).

---

[4]We omitted the Risk Avoidance scale because the scale reliability was questionable in its original version (for a discussion, see also Vergara *et al.* [29]).

**Table 2.** This table displays extracted predictors differing from random noise for the *a*, *b*, *c* path of the mediation Thermoregulation -> Attachment -> Personality models. Full table including full conditional random forests results is posted on our OSF page.

| | variables to be predicted | extracted thermoregulation predictors (which probably differ from random noise) | extracted attachment predictors (which probably differ from random noise) |
|---|---|---|---|
| mediators | Anxiety 'Modified' | solitary thermoregulation | |
| | Avoidance 'Modified' | social thermoregulation | |
| dependant variables | well-being | none | Anxiety 'Modified' |
| | anxiousness | solitary thermoregulation | Anxiety 'Modified' |
| | self-esteem | none | Anxiety 'Modified' & Avoidance 'Modified' |
| | self-discipline | none | none |
| | Loneliness 'Modified' | social thermoregulation | Anxiety 'Modified' & Avoidance 'Modified' |
| | Social Dominance Orientation 'Modified' | solitary thermoregulation | none |
| | right prejudice | none | Avoidance 'Modified' |
| | impulsiveness | none | Anxiety 'Modified' |
| | reflection | none | none |
| | trust | solitary thermoregulation and social thermoregulation | Anxiety 'Modified' & Avoidance 'Modified' |
| | Right Wing Authoritarianism 'Modified' | none | Avoidance 'Modified' |
| | stimulation | none | none |
| | sociability | social thermoregulation | Anxiety 'Modified' & Avoidance 'Modified' |
| | leadership | none | Anxiety 'Modified' & Avoidance 'Modified' |
| | empathy | social thermoregulation and solitary thermoregulation | Avoidance 'Modified' |

We again only summarize the results here. The extracted variables for each of our 100 conditional random forests are displayed in the second section of table 2. For Well-being, Anxiousness, and Impulsiveness, the only attachment variable predicting these personality factors beyond random noise was Attachment Anxiety 'Modified'. In other words, when trying to predict these three variables one should use Attachment Anxiety as mediator.

For Self-esteem, Loneliness 'Modified', Trust, Sociability, and Leadership the variables predicting beyond random noise were Attachment Anxiety 'Modified' followed by Attachment Avoidance 'Modified'. Both Attachment Anxiety 'Modified' and Avoidance 'Modified' can thus be used as mediator when predicting these variables. For Right Prejudice, Right Wing Authoritarianism 'Modified', and Empathy the only attachment variable predicting beyond random noise was Attachment Avoidance 'Modified'. Only Attachment Avoidance 'Modified' should thus be used as a predictor in the mediation analyses. For the other personality variables, none of the attachment variables were predictors beyond random noise.

## 4.3. STRAQ-1 predicting personality (mediation path *c*)

For the dependent variables (personality), we ran 100 conditional random forests per personality outcome variable with the STRAQ-1 variables (excluding the Risk Avoidance scale) as predictors. We

again inserted other personality variables as predictors of non-interest. We summarize the main findings here; the extracted variables for each of our 100 conditional random forests are again displayed in the third section of table 2.

For Anxiousness and Social Dominance Orientation 'Modified' the only thermoregulation variable predicting these personality factors beyond random noise was Solitary Thermoregulation. Solitary Thermoregulation should thus be used as a predictor for these two variables. For Loneliness and Sociability, the only thermoregulation variable predicting these personality factors beyond random noise was Social Thermoregulation. Social Thermoregulation should thus be used as predictor for these two variables. For Trust and Empathy, the thermoregulation variables predicting these personality factors beyond random noise were Social Thermoregulation first and Solitary Thermoregulation second. Both Social and Solitary Thermoregulation should thus be used as predictors for Trust and Empathy. For the other personality variables, none of the thermoregulation variables were predictors beyond the random noise.

We have now established some (seemingly robust) relationships for the different paths of the mediation. However, our conditional random forests also presented some surprises, as not all relationships between (Social) Thermoregulation and personality were mediated by attachment. In our next step, we selected the personality factors that were candidates for prediction for our thermoregulated-related variables (with corresponding mediators, if applicable). We display a summary of the conditional random forest results and consistent mediation models in table 3.

## 4.4. Relationships between the personality factors and their predictors

Relationships between all the variables of the mediation models are displayed in table 3. More specifically, we examined the relationships between the thermoregulation variables and their attachment mediators, the relationships between the attachment mediators and the selected personality factors, and the relationships between the thermoregulation variables and the selected personality factors. To examine whether these were linear or not, we compared the traditional regression lines with the locally estimated scatterplot smoothing (LOESS) curves by calculating as many local least-squares regression functions as there are data segments. All scatter plots are available on our OSF page (https://osf.io/74fr3/). By comparing the LOESS curves with the regression lines, we found that none of the relations between the variables of our mediation models were nonlinear. We thus further proceeded with traditional mediation analyses using linear regression.

# 5. Regression analyses

Based on our conditional random forest results, we continued to explore more traditional regression analyses. This included a hypothesis-generating mediation model. Conducting these analyses would also allow us to write down the prediction for our confirmatory study, as well as better understand the relationships between variables (in addition, our conditional random forests included variables of non-interest and may thus not have represented the most optimal relationship between variables). We split these up into mediation analyses first and simple correlations second.

## 5.1. Mediation models

To test the indirect effect in the mediations, we relied on the joint significance approach. This approach consists of testing the significance of the two components constituting the indirect effect, which is the relationship between the independent variable and the mediator (*a* path) and the relationship between the mediator and the dependent variable (*b* path) controlling for the independent variable (figure 1). The indirect effect is the product of those two coefficients and is significant only when both the *a* path and the *b* path are significant. This method has a lower Type 1 error rate than methods that rely on bootstrapping [72].

We also used the Monte Carlo method as a complementary approach to estimate the significance of the indirect effect and the confidence interval. The Monte Carlo method first estimates the standard error distributions for *a* and *b* for the sample and then computes 5000 random samples based on the product of those two distributions estimates. We report the results of the joint significance test and the Monte Carlo method in table 4.

**Table 3.** Summary of exploratory results and projected models for regression analyses.

| dependant variable | mediator | predictor(s) | resulting model |
|---|---|---|---|
| well-being | n.a. | Anxiety 'Modified' | correlation |
| anxiousness | Anxiety 'Modified' | Solitary Thermoregulation | mediation (Solitary Thermoregulation -> Anxiety -> Anxiousness) |
| self-esteem | n.a. | Anxiety 'Modified' and Avoidance 'Modified' | correlation |
| self-discipline | n.a. | n.a. | none |
| Loneliness 'Modified' | Anxiety 'Modified'; Avoidance 'Modified' | Social Thermoregulation | mediation (Social Thermoregulation -> Avoidance -> Loneliness 'Modified') |
| Social Dominance Orientation 'Modified' | n.a. | Solitary Thermoregulation | correlation |
| right prejudice | n.a. | Avoidance 'Modified' | correlation |
| impulsiveness | n.a. | Anxiety 'Modified' | correlation |
| reflection | n.a. | n.a. | none |
| trust | Anxiety 'Modified'; Avoidance 'Modified' | Solitary Thermoregulation and Social Thermoregulation | mediation (Solitary Thermoregulation -> Anxiety ->Trust and Social Thermoregulation -> Avoidance -> Trust) |
| Right Wing Authoritarianism 'Modified' | n.a. | Avoidance 'Modified' | correlation |
| stimulation | n.a. | n.a. | none |
| sociability | Anxiety 'Modified'; Avoidance 'Modified' | Social Thermoregulation | mediation (Social Thermoregulation -> Avoidance -> Sociability) |
| leadership | n.a. | Anxiety 'Modified' and Avoidance 'Modified' | Correlation |
| empathy | Avoidance 'Modified' | Social Thermoregulation | mediation (Social Thermoregulation -> Avoidance -> Empathy) |

For all mediation models, we decided to include the sex as control variable as per the recommendation by IJzerman *et al.* [73]. They found that participants' sex may have an influence on the effect size of variables related to social thermoregulation (we did not make any further inferences about sex, because our sample was not representative of men versus women). We include the analyses without sex as control variable on our OSF page (https://osf.io/f6qun/).[5]

We found that the effect of Solitary Thermoregulation on Anxiousness was mediated by Attachment Anxiety 'Modified' and that the effect of Solitary Thermoregulation on Trust was mediated by

---

[5]In half of our sample, the correlation between solitary thermoregulation and sex was $r = 0.31$, $p < 0.001$ and the correlation between social thermoregulation and sex was $r = 0.04$, $p = 0.44$. Furthermore, Vergara *et al.* [29] found a link between thermoregulation variables and the relationship status and we might expect similar relation in our sample as well. In half of our sample, the correlation between solitary thermoregulation and the relationship status was $r = -0.11$, $p = 0.09$ and the correlation between social thermoregulation and the relationship status was $r = 0.22$, $p < 0.001$.

**Table 4.** Exploratory mediation analyses for the Thermoregulation -> Attachment -> Personality models. *Note.* Coefficients *b* are unstandardized regression coefficients. $*p < 0.05$, $**p < 0.01$, $***p < 0.001$.

| variables | | | *a* path | | | *b* path | | | *c* path | | | *c'* path | | | | indirect effect |
|---|---|---|---|---|---|---|---|---|---|---|---|---|---|---|---|---|
| independant variable | mediator | dependant variable | *b* | *t* | *p* | *b* | *t* | *p* | *b* | *t* | *p* | *b* | *T* | *p* | | Monte Carlo 95% CI |
| Solitary Thermoregulation | Anxiety 'Modified' | anxiousness | 0.30** | $t_{242} = 2.79$ | <0.01 | 0.27*** | $t_{238} = 7.65$ | <0.01 | 0.19** | $t_{244} = 2.87$ | <0.01 | 0.13* | $t_{238} = 2.24$ | 0.03 | | [0.02; 0.15] |
| | | trust | 0.30** | $t_{242} = 2.79$ | <0.01 | −0.23*** | $t_{239} = 5.35$ | <0.01 | −0.16* | $t_{245} = 2.18$ | 0.03 | −0.13 | $t_{239} = 1.74$ | 0.08 | | [−0.13; −0.02] |
| Social Thermoregulation | Avoidance 'Modified' | trust | −0.56*** | $t_{240} = 7.42$ | <0.01 | −0.17*** | $t_{237} = 3.50$ | <0.01 | 0.20*** | $t_{247} = 3.73$ | <0.01 | 0.10 | $t_{237} = 1.58$ | 0.12 | | [0.04; 0.16] |
| | | Loneliness 'Modified' | −0.56*** | $t_{240} = 7.42$ | <0.01 | −0.16*** | $t_{236} = 5.00$ | <0.01 | 0.18*** | $t_{245} = 4.83$ | <0.01 | 0.07 | $t_{236} = 1.75$ | 0.08 | | [0.05; 0.14] |
| | | sociability | −0.56*** | $t_{240} = 7.42$ | <0.01 | −0.12* | $t_{234} = 2.39$ | 0.02 | 0.22*** | $t_{244} = 3.84$ | <0.01 | 0.16* | $t_{234} = 2.41$ | 0.02 | | [0.01; 0.13] |
| | | empathy | −0.56*** | $t_{240} = 7.42$ | <0.01 | −0.12*** | $t_{238} = 3.76$ | <0.01 | 0.14*** | $t_{248} = 3.57$ | <0.01 | 0.06 | $t_{238} = 1.32$ | 0.19 | | [0.03; 0.12] |

**Table 5.** Pearson's correlations between thermoregulation variables and personality variables. $^*p < 0.05$, $^{**}p < 0.01$, $^{***}p < 0.001$.

| independent variable | dependent variable | Pearson's correlation | t | p |
|---|---|---|---|---|
| Solitary Thermoregulation | Social Dominance Orientation 'Modified' | 0.13$^*$ | $t_{244} = 1.98$ | 0.05 |
| | Empathy | 0.17$^{**}$ | $t_{246} = 2.75$ | <0.01 |
| Social Thermoregulation | Sociability | 0.24$^{***}$ | $t_{245} = 3.85$ | <0.01 |

Attachment Anxiety 'Modified'. We also found that the effect of Social Thermoregulation on Trust, Loneliness 'Modified', and Empathy was mediated by Attachment Avoidance 'Modified' and that the effect of Social Thermoregulation on Sociability was mediated by Attachment Avoidance 'Modified'. The results of mediation analyses are displayed in table 4.

We then proceeded with correlational analyses between the identified personality factors and thermoregulation variables. Indeed, based on conditional random forest (CRF) results, not all relationships seemed to be mediated by attachment. In this additional set of analyses, we found that Social Dominance Orientation 'Modified' and Empathy was predicted by Solitary Thermoregulation and that Sociability was predicted by Social Thermoregulation (note that Attachment Avoidance was very close to random noise when predicting Sociability in our conditional random forests, so we did not presume mediation). Note that the relationship between Solitary Thermoregulation and Social Dominance Orientation 'Modified' is weak, unlikely to replicate, and must be interpreted with caution. Pearson's correlations and their $p$-values are displayed in table 5.

## 5.2. Robustness analyses

To ensure the robustness of our analyses, we ran auxiliary analyses of the attachment factors as they are traditionally envisioned by the original scale's creators (for the Big Five and the ECR). This thus provides a robustness check for our results. For the original Agreeableness scale we found comparable results as for the modified scales Trust and Empathy (i.e. prediction by Social Thermoregulation and mediation by Avoidance). For the original Neuroticism scale, we found comparable results as for the modified scale Anxiousness (i.e. prediction by Solitary Thermoregulation and mediation by Anxiety). For the original Extraversion scale, we found comparable results as for the modified scales Loneliness 'Modified' and Sociability (i.e. prediction by Social Thermoregulation and mediation by Avoidance). Again, we controlled for the sex in all mediation models. The results of mediation analyses with traditional factors are displayed in table 6.

# 6. Confirmatory results

## 6.1. Conditional random forests

We had two confirmatory sets. Our first confirmatory set consisted of the second half of the first sample. This confirmatory set allowed us to test our predictions, and, in case our model is inaccurate, adjust before we tested in the second dataset. We tested our predictions by evaluating the replicability of each conditional random forests using the predict function. To estimate the replicability of conditional random forests models in the test dataset, we then calculated the squared correlation between these predictions and the actual values in the test dataset.

As each conditional random forest model was computed 100 times, we computed 100 squared correlations for each model prediction that were then averaged out, resulting in one squared correlation for each conditional random forest model. To evaluate the overall replicability of the conditional random forest models, we calculated the mean squared correlation resulting from the average of all squared correlations associated with each conditional random forest model. The $M_{R\text{squared}}$ was 0.35 ($s.d._{R\text{squared}} = 0.13$, $\text{Min}_{R\text{squared}} = 0.11$, $\text{Max}_{R\text{squared}} = 0.65$).

To estimate the replicability of the conditional random forest models used to create each mediation model, we then calculated the mean squared correlation for each mediation model by averaging the three squared correlations associated with CRF models used to create the three paths of the mediation

**Table 6.** Exploratory mediation analyses for the Thermoregulation -> Attachment -> Big Five models. *Note.* Coefficients *b* are unstandardized regression coefficients. $^* p < 0.05$, $^{**} p < 0.01$, $^{***} p < 0.001$.

| variables | | | a path | | | b path | | | c path | | | c' path | | | Indirect effect |
|---|---|---|---|---|---|---|---|---|---|---|---|---|---|---|---|
| independant variable | mediator | dependant variable | b | t | p | b | t | p | b | t | p | b | T | p | Monte Carlo 95% CI |
| Solitary Thermoregulation | anxiety | neuroticsm | 0.28** | $t_{239} = 2.63$ | <0.01 | 0.29*** | $t_{234} = 9.35$ | <0.01 | 0.12* | $t_{243} = 2.11$ | 0.04 | 0.04 | $t_{234} = 0.73$ | 0.47 | [0.02; 0.15] |
| | | agreableness | 0.28** | $t_{239} = 2.63$ | <0.01 | −0.14*** | $t_{230} = 5.33$ | <0.01 | 0.02 | $t_{239} = 0.40$ | 0.69 | 0.04 | $t_{230} = 0.95$ | 0.34 | [−0.07; −0.01] |
| Social Thermoregulation | avoidance | agreableness | −0.56*** | $t_{240} = 7.42$ | <0.01 | −0.11*** | $t_{232} = 4.28$ | <0.01 | 0.10*** | $t_{242} = 3.28$ | <0.01 | 0.03 | $t_{232} = 0.86$ | 0.39 | [0.03; 0.10] |
| | | extraversion | −0.56*** | $t_{240} = 7.42$ | <0.01 | −0.06 | $t_{233} = 1.92$ | 0.06 | 0.20*** | $t_{242} = 5.14$ | <0.01 | 0.16*** | $t_{233} = 3.62$ | <0.01 | [0.01; 0.07] |

**Table 7.** Summary of the replication of the *a*, *b*, and *c* paths of the Thermoregulation -> Attachment -> Personality models and mean squared correlations summarizing the replicability of CRF models used to create these mediation models.

| mediation model | Mean R squared (s.d., Min, Max) | not replicated (a,b,c) | weak (a, b,c) | medium (a,b,c) | strong (a,b,c) | overall replication |
|---|---|---|---|---|---|---|
| Solitary Thermoregulation -> Anxiety 'Modified' -> Anxiousness | M = 0.44 (s.d. = 0.12, Min = 0.35, Max = 0.58) | −, −, − | +, +, + | +, +, + | −,−,− | medium replication |
| Solitary Thermoregulation -> Anxiety 'Modified' -> Trust | M = 0.29 (s.d. = 0.08, Min = 0.2, Max = 0.35) | −, −, + | +, +, − | +, +, − | −,−,− | not replicated |
| Social Thermoregulation -> Avoidance 'Modified' -> Trust | M = 0.28 (s.d. = 0.07, Min = 0.20, Max = 0.33) | −, −, + | +, +, − | +, +, − | −,−,− | not replicated |
| Social Thermoregulation -> Avoidance 'Modified' -> Loneliness 'Modified' | M = 0.40 (s.d. = 0.06, Min = 0.33, Max = 0.44) | −, −, − | +, +, + | +, +, + | −,−,− | medium replication |
| Social Thermoregulation -> Avoidance 'Modified' -> Sociability | M = 0.33 (s.d. = 0.01, Min = 0.32, Max = 0.34) | −, +, − | +, −, + | +, −, + | −,−,− | not replicated |
| Social Thermoregulation -> Avoidance 'Modified' -> Empathy | M = 0.29 (s.d. = 0.04, Min = 0.26, Max = 0.33) | −, −, − | +, +, + | +, +, + | −,−,− | medium replication |

(*a*, *b*, and *c* paths). We provide the mean squared correlation for each mediation model in the left panel of table 7. Overall, these squared correlations revealed a medium fit between predictions from the conditional random forest models generated in training data and actual test data. Therefore, even though this reveals sufficient replicability of conditional random forest models in the test data, we cannot exclude the possibility that we may have overfitted our models to the data and thus made the wrong assumptions about the mediation models. By subsequently replicating the mediation models, we should be able to shed further light about the robustness of our models.

## 6.2. Regression models

We then tested how accurate our predictions were for our regression analyses. We relied on formalized predictions from our regression models generated in the exploratory phase in our training data (see also [74]). We display these formalized predictions in table 8.

## 6.3. Criteria for replication

We set our replication criteria *a priori*: to determine the replication of the exploratory results, we could examine whether the beta estimates of the three paths of the mediation (*a*, *b*, and *c* paths) were

**Table 8.** Formalized predictions.

| variables | | | | | | |
| --- | --- | --- | --- | --- | --- | --- |
| independent variable | mediator | dependant variable | *a* path | *b* path | *c* path | *c′* path |
| Solitary Thermoreglation | Anxiety 'Modified' | Anxiousness | Anxiety 'Modified' = 2.72 + 0.30 Solitary Thermoregulation + 0.18 Sex | Anxiousness = 1.15 + 0.27 Anxiety 'Modified' + 0.13 Solitary Thermoregulation + 0.27 Sex | Anxiousness = 1.99 + 0.19 Solitary Thermoregulation + 0.33 Sex | Anxiousness = 1.15 + 0.13 Solitary Thermoregulation + 0.27 Anxiety 'Modified' + 0.27 Sex |
| | | Trust | | Trust = 4.24 − 0.23 Anxiety 'Modified' − 0.13 Solitary Thermoregulation + 0.07 Sex | Trust = 3.48 − 0.16 Solitary Thermoregulation + 0.02 Sex | Trust = 4.24 − 0.13 Solitary Thermoregulation − 0.23 Anxiety 'Modified' + 0.07 Sex |
| Social Thermoregulation | Avoidance 'Modified' | Trust | Avoidance 'Modified' = 4.50 − 0.56 Social Thermoregulation + 0.07 Sex | Trust = 3.12 − 0.17 Avoidance 'Modified' + 0.10 Social Thermoregulation − 0.08 Sex | Trust = 2.35 + 0.20 Social Thermoregulation − 0.11 Sex | Trust = 3.12 + 0.10 Social Thermoregulation − 0.17 Avoidance 'Modified' − 0.08 Sex |
| | | Loneliness 'Modified' | | Loneliness 'Modified' = 3.31 − 0.16 Avoidance 'Modified' + 0.07 Social Thermoregulation + 0.03 Sex | Loneliness 'Modified' = 2.52 + 0.18 Social Thermoregulation + 0.06 Sex | Loneliness 'Modified' = 3.31 + 0.07 Social Thermoregulation − 0.16 Avoidance 'Modified' + 0.03 Sex |
| | | Sociability | | Sociability = 2.96 − 0.12 Avoidance 'Modified' + 0.16 Social Thermoregulation − 0.08 Sex | Sociability = 2.42 + 0.22 Social Thermoregulation + 0.02 Sex | Sociability = 2.96 + 0.16 Social Thermoregulation − 0.12 Avoidance 'Modified' − 0.08 Sex |
| | | Empathy | | Empathy = 4.47 − 0.12 Avoidance 'Modified' + 0.06 Social Thermoregulation + 0.31 Sex | Empathy = 3.88 + 0.14 Social Thermoregulation + 0.30 Sex | Empathy = 4.47 + 0.06 Social Thermoregulation − 0.12 Avoidance 'Modified' + 0.31 Sex |

significant and we could examine whether the direction of the three paths (i.e. negative or positive) were identical to those in the exploratory results.

If the result were in the same direction, we would consider it a weak replication. For a medium test of the prediction, we would rely on the lower bound of the confidence intervals of the beta estimates obtained in the exploratory phase to define a minimum beta interest to be replicated in the confirmatory phase. For a strong test of the prediction, we would also test the difference between the estimate in the exploratory model and in the confirmatory models through Z-tests. Only a significant difference resulting from the superiority of the confirmatory beta estimate over exploratory beta estimate was considered as a strong replication. Because this point prediction replication approach leaves us very little margin for estimation error, we expected our replication rate to be very low for the strong test of our prediction.

A summary of the replication of the regression analyses for each path of mediation models ($a$, $b$, and $c$ paths) is displayed in the middle panel of table 7. Of the 18 paths analysed through regression analyses in the confirmatory dataset, only three were considered as not replicated at all, 15/18 were replicated according to our weak criteria, 15/18 according to our medium criteria, and 0/18 according to our strong criteria.

As all three pathways of the mediation needed to be replicated for the meditation to be replicated, three out of six mediations were considered replicated. Specifically, we replicated the effect of Solitary Thermoregulation on Anxiousness, with mediator Attachment Anxiety 'Modified'. We also replicated the effect of Social Thermoregulation on Loneliness 'Modified', with as mediator Attachment Avoidance 'Modified'. We further replicated the effect of Social Thermoregulation on Empathy, with as mediator Attachment Avoidance 'Modified'. Other mediation models were considered as not replicated due to one path not being statistically significant. The results of the confirmatory mediation analyses are displayed in table 9.

# 7. Discussion study 1

In Study 1, we investigated the link between social thermoregulation and personality. We relied on an exploratory approach consisting of splitting our dataset in a training and a testing dataset. Because we conducted our study in a sample (French students in Grenoble) where little validation had been done, we first conducted exploratory factor analyses to create personality factors on multiple individual difference scales. We then investigated the relations between these new personality factors, (only somewhat) modified attachment variables, and social thermoregulation variables through machine learning. Based on machine learning results, we created mediation models that were first analysed in the first half of the data.

These analyses revealed that of the six mediation models tested, all were significant. Accordingly, we found that Attachment Anxiety 'Modified' mediated the relationship between Solitary Thermoregulation and Anxiousness as well as the relationship between Solitary Thermoregulation and Trust. Moreover, we also found that Attachment Avoidance 'Modified' mediated the relation between Social Thermoregulation and Trust, Social Thermoregulation and Loneliness 'Modified', Social Thermoregulation and Sociability, and Social Thermoregulation and Empathy.

The Stage 1 manuscript associated with this Registered Report was granted in-principle acceptance on 20 November 2020. The accepted Stage 1 manuscript, unchanged from the point of in-principle acceptance, may be viewed at https://osf.io/b8jq6/. Then, after in-principle acceptance from this journal, we replicated these mediation models in the testing set of the data. Before moving to this step, we evaluated the replicability of our conditional random forests in the test dataset by calculating the squared correlation between predictions from CRF models computed in the training and actual testing data. Overall, these squared correlations revealed medium fit between CRF model predictions and actual test data. As our mediation models were created based on CRF results, medium fit between CRF models and test data does not totally rule out the possibility that we may have made wrong mediation assumptions due to CRF overfitting.

We therefore then moved on to testing the replicability of our mediation models in the test dataset to investigate this possibility. We found that 15/18 of the effects were replicated, but only three out of the six mediation models tested could be considered as medium replications (i.e. their confidence interval was overlapping and they were not only significant at the same level): the mediation between Solitary Thermoregulation, Attachment Anxiety 'Modified', and Anxiousness; the mediation between Social Thermoregulation, Attachment Avoidance 'Modified', and Loneliness 'Modified'; the mediation

**Table 9.** Confirmatory mediation analyses for the Thermoregulation -> Attachment -> Personality models. *Note.* Coefficients *b* are unstandardized regression coefficients. *$p < 0.05$, **$p < 0.01$, ***$p < 0.0$.

| Variables | | | *a* path | | | *b* path | | | *c* path | | | *c'* path | | | Indirect effect |
| --- | --- | --- | --- | --- | --- | --- | --- | --- | --- | --- | --- | --- | --- | --- | --- |
| independant variable | mediator | dependant variable | *b* | *t* | *p* | *b* | *t* | *p* | *b* | *t* | *p* | *b* | *t* | *p* | Monte Carlo 95% CI |
| Solitary | Anxiety 'Modified' | anxiousness | 0.27* | $t_{240} = 2.50$ | 0.01 | 0.29*** | $t_{232} = 8.21$ | <0.01 | 0.25*** | $t_{235} = 3.67$ | <0.01 | 0.16** | $t_{232} = 2.7$ | <0.01 | [0.02; 0.14] |
| Thermoreglation | | trust | 0.27* | $t_{240} = 2.50$ | 0.01 | −0.24*** | $t_{238} = 5.33$ | <0.01 | −0.02 | $t_{242} = 0.20$ | 0.84 | 0.05 | $t_{238} = 0.70$ | 0.48 | [−0.12; −0.01] |
| Social Thermoregulation | Avoidance 'Modified' | trust | −0.44*** | $t_{236} = 6.00$ | <0.01 | −0.21*** | $t_{234} = 4.52$ | 0.14 | 0.08 | $t_{243} = 1.50$ | 0.13 | −0.04 | $t_{234} = 0.71$ | 0.48 | [0.05; 0.15] |
| | | Loneliness 'Modified' | −0.44*** | $t_{236} = 6.00$ | <0.01 | −0.20*** | $t_{230} = 6.00$ | <0.01 | 0.12** | $t_{239} = 2.99$ | <0.01 | 0.02 | $t_{230} = 0.40$ | 0.69 | [0.05; 0.13] |
| | | sociability | −0.44*** | $t_{236} = 6.00$ | <0.01 | −0.08 | $t_{231} = 1.65$ | 0.1 | 0.32*** | $t_{240} = 5.61$ | <0.01 | 0.28*** | $t_{231} = 4.57$ | <0.01 | [−0.01; 0.09] |
| | | empathy | −0.44*** | $t_{232} = 6.00$ | <0.01 | −0.11** | $t_{232} = 3.32$ | <0.01 | 0.11** | $t_{241} = 2.93$ | <0.01 | 0.06 | $t_{232} = 1.59$ | 0.11 | [0.02; 0.08] |

between Social Thermoregulation, Attachment Avoidance 'Modified', and Empathy. Other mediation models were considered as not replicated because for all of them one path was not significant.

To further increase confidence in our findings, we then aimed to refine our predictions based on confirmatory results and test them in an out-of-sample dataset in a second study. Specifically, we would only replicate mediation models that were at least considered as weak replications. Thus, we aimed to replicate three of our six mediation models. However, from Study 1 to 2, not all items were asked, as researchers that selected scales for the testweek elected to include different scales. The differences between Studies 1 and 2 are displayed in appendix C. The mediations that included different items were Solitary Thermoregulation onto Anxiousness, mediated by Anxiety 'Modified' and Social Thermoregulation onto Loneliness 'Modified' mediated by Avoidance 'Modified'. When we re-ran the mediations for those models, we again found them to replicate ($p$s for all paths except the $c'$ path less than 0.01; see also https://osf.io/42xcg/). Therefore, for an out-of-sample dataset, we predicted the same relationships as we replicated in our testing dataset from Study 1, despite the differences in items. In Study 2, we will restrict ourselves to testing the mediation models.

# 8. Study 2

## 8.1. Participants

Participants ($N = 334$) were recruited from Université Grenoble Alpes in France in 2019. Psychology students (41 men, 283 women, 2 others, and 8 missing) answered a battery of online questionnaires on Qualtrics. They were 90 in L1, 104 in L2, 106 in L3, 25 in M1, and 1 in M2 (8 missing). The $M_{age}$ was 20.36 ($s.d._{age} = 3.10$), $M_{self-reported\ height}$ was 166.87 ($s.d._{self-reported\ height} = 8.66$)[6] and the $M_{self-reported\ weight}$ was 61.26 ($s.d._{self-reported\ weight} = 13.15$). Amongst our participants 77 were smokers and 248 were not (9 missing). Also, 183 were in a relationship, 129 were not, and 14 preferred not to answer.

## 8.2. Confirmatory results

### 8.2.1. Regression models

In Study 2, we relied on formalized predictions resulting from regression analyses in the whole Study 1 dataset (i.e. merged train and test datasets). We computed these predictions with identical factors to those in Study 2 (i.e. without items that were missing for the factors Anxiousness and Loneliness 'Modified') so that predictions reflect what could reasonably be expected for Study 2. These formalized predictions may be found at https://osf.io/cmbdy/.

### 8.2.2. Criteria for replication

We applied the same replication criteria in Study 2 as for Study 1, with the note that we compared the coefficients from the ones obtained through mediation analyses from the entire Study 1 dataset. A summary of the replication of the regression analyses for each path of mediation models ($a$, $b$, and $c$ paths) is displayed in table 10. Of the nine paths analysed through regression analyses in the Study 2 dataset, three were considered as not replicated at all, 6/9 were replicated according to our weak criteria, 6/9 according to our medium criteria, and 0/9 according to our strong criteria.

As all three pathways of the mediation needed to be replicated for the mediation to be replicated, only one mediation model was considered as fully replicated in Study 2: we replicated the effect of Social Thermoregulation on Loneliness 'Modified', with as mediator Attachment Avoidance 'Modified'. Other mediation models were considered as not replicated due to at least one path not being statistically significant. The results of the confirmatory mediation analyses are displayed in table 11.

# 9. Discussion study 2

In Study 2, we aimed to test Study 1 predictions in an out-of-sample dataset based on regression analyses generated from the entirety of Study 1 and preregistered these predictions on the OSF. Mediation

---

[6]We again removed observations under 100 cm for the calculation of the $M_{self-reported\ height}$ and $s.d._{self-reported\ height}$, given the high unlikelihood of these scores (and were probably in our data due to typing errors).

**Table 10.** Summary replication Study 2. This table displays the nature of the replication for each path of the mediation (a, b, c). This table displays the nature of the replication for each path of the mediation (*a*, *b*, *c*). '+' means that the path meets the criteria of the corresponding replication type while '−' means that the path does not meet the criteria of the corresponding replication type. Mean squared correlations for each mediation model were computed by averaging the three squared correlations associated with the three CRF models used to create each path of the mediation.

| mediation model | not replicated (*a,b,c*) | weak (*a,b,c*) | medium (*a,b,c*) | strong (*a,b,c*) | overall replication |
|---|---|---|---|---|---|
| Solitary Thermoregulation -> Anxiety 'Modified' -> Anxiousness | +, −, + | −, +, − | −, +, − | −,−,− | not replicated |
| Social Thermoregulation -> Avoidance 'Modified' -> Loneliness 'Modified' | −, −, − | +, +, + | +, +, + | −,−,− | medium replication |
| Social Thermoregulation -> Avoidance 'Modified' -> Empathy | −, −, + | +, +, - | +, +, − | −,−,− | not replicated |

analyses revealed that 6/9 of the effects were replicated but only one out of three mediations could be considered as a replication: we replicated the effect of Social Thermoregulation on Loneliness 'Modified' mediated by Attachment Avoidance 'Modified'. Other models were not replicated due to at least one path not being statistically significant.

As an aside, the effect of Solitary Thermoregulation on the mediator Attachment Anxiety 'Modified' did not replicate in Study 2 even though this relation was found in both Study 1 datasets. A rationale for this could be that we may have lacked power to detect this link. Indeed, in both Studies 1 and 2, this relation had a smaller effect size than the link between Social Thermoregulation and Attachment Avoidance 'Modified' (which replicated in all sets of analyses). This is also consistent with Vergara *et al.* [29] who found the link between Social Thermoregulation and Attachment Avoidance to be stronger than the link between Solitary Thermoregulation and Attachment Anxiety[7]. To indeed understand whether power was a factor in non-replication, we added exploratory analyses where we merged Studies 1 and 2.

# 10. Studies 1 and 2 exploratory results

As these results are exploratory, they can only be used to understand why we did not replicate our analyses elsewhere and they can potentially be used for hypothesis generation, but not confirmation. To better understand why our results did not replicate, we ran the six mediation models that we had initially detected via conditional random forests in the first half of Study 1 in the entire dataset (i.e. the merged Studies 1 and 2 datasets with $N = 844$) with models applied in Study 2 (i.e. without the missing items due to selections made by other researchers). While in the split approach, only one out of six replicated throughout datasets, when we ran the analyses in the entire dataset, five had three paths that were statistically significant. More specifically, (1) Attachment Anxiety 'Modified' mediated the relations between Solitary Thermoregulation and Anxiousness and (2a) Attachment Avoidance 'Modified' mediated the relation between Social Thermoregulation and Trust, (2b) Social Thermoregulation and Loneliness 'Modified', (2c) Social Thermoregulation and Sociability, and (2d) Social Thermoregulation and Empathy. The reader may find the exploratory mediation results of Studies 1 and 2 at https://osf.io/6yev2/.

# 11. General discussion

Altogether, in these two studies we explored the link between (social) thermoregulation on one hand and personality on the other hand and the link is mediated by attachment. Using a conservative cross-

[7]In Vergara *et al.* [29] Social Thermoregulation and Attachment Avoidance, Pearson's correlation was $r = -0.32^{***}$, whereas Pearson's correlation for Solitary Thermoregulation and Attachment Anxiety was $r = 0.08^{**}$.

**Table 11.** Study 2 mediation analyses for the Thermoregulation -> Attachment -> Personality models. *Note.* Coefficients $b$ are unstandardized regression coefficients. $*p < 0.05$, $**p < 0.01$, $***p < 0.001$.

| variables | | | *a* path | | | *b* path | | | *c* path | | | *c'* path | | | Indirect effect |
|---|---|---|---|---|---|---|---|---|---|---|---|---|---|---|---|
| independant variable | mediator | dependant variable | b | t | p | b | t | p | b | t | p | b | t | p | Monte Carlo 95% CI |
| Solitary Thermoregulation | Anxiety 'Modified' | anxiousness | 0.08 | $t_{249} = 0.76$ | 0.45 | 0.26*** | $t_{193} = 6.00$ | <0.01 | 0.08 | $t_{224} = 1.14$ | 0.25 | 0.10 | $t_{193} = 1.32$ | 0.19 | [−0.03; 0.08] |
| Social Thermoregulation | Avoidance 'Modified' | Loneliness 'Modified' | −0.45*** | $t_{245} = 5.45$ | <0.01 | −0.15*** | $t_{263} = 4.36$ | <0.01 | 0.14*** | $t_{263} = 3.56$ | <0.01 | 0.06 | $t_{219} = 1.39$ | 0.17 | [0.03; 0.11] |
| | | empathy | −0.45*** | $t_{245} = 5.45$ | <0.01 | −0.14*** | $t_{200} = 4.45$ | <0.01 | 0.06 | $t_{236} = 1.64$ | 0.10 | 0.00 | $t_{200} = 0.18$ | 0.86 | [0.03; 0.11] |

Coefficients $b$ are unstandardized regression coefficients.

$*p < 0.05$, $**p < 0.01$, $***p < 0.001$.

validation approach and through out-of-sample testing, the only effect that replicated throughout all datasets was the relationship between social thermoregulation desires and feelings of loneliness, which was mediated by attachment avoidance.

## 11.1. Why does (social) thermoregulation predict little beyond loneliness?

*A priori*, we had anticipated that (social) thermoregulation would be formative for personality on facets other than loneliness and attachment avoidance. However, we found little evidence to substantiate that view. Why could this be? The first and most obvious answer is that no such link exists. We consider that a reasonable possibility. However, we would like to point to some of the shortcomings in our research that could limit the extent to which that answer is certain.

Indeed, and second, our research may not have been sufficiently powered to detect these effects. In our first dataset, for instance, we did find effects on the factor sociability. To explore whether the small rate of replication was due to a lack of power, we conducted exploratory analyses for the six models analysed in Study 1 on an entire dataset created by merging the Studies 1 and 2 datasets. These analyses revealed that on the six mediation models tested, five were considered as statistically significant; that is, 11/12 of the effects tested were significant in our merged sample.

We then ran sensitivity analyses to understand what the minimum effect sizes we could have expected to find in Study 1 and Study 2 datasets. When we took the effect sizes from the merged dataset and conducted a sensitivity analysis to understand whether we would have been able to observe them in the separate Studies 1 and 2, the effect sizes were smaller than the minimum effect size boundary (https://osf.io/zj75c/; beyond a potential cultural variation between the French and other samples, this also provides a rationale for the non-replication of the link between Solitary Thermoregulation and Attachment Anxiety in Study 2, which has been found in two previous sets of analyses and in another study; [29]). Although such effects may thus replicate in larger samples, the alternative conclusion is that these effects are simply too small to care about and to invest resources into for potential interventions.

And yet, there is a third answer to why the relationship between (social) thermoregulation and personality are not detected as predicted. The questions we ask are somewhat constrained due to our measurement. When we assess social thermoregulation, we assess people's desires to socially thermoregulate, not necessarily the degree to which people know their confidence that others will be available (a crucial component when thinking about the reliability and predictability of the environment). Because of the way we had developed our measurement, the extent to which the concept of social thermoregulation maps onto the measure is thus limited. We are trying to address such limitations in a follow-up project to better measure social thermoregulation [75].

## 11.2. Loneliness in the elderly and around the world: constraints on generality [76]

Although we feel comfortable making strong assumptions about the statistical link between social thermoregulation and loneliness, we feel that such assumptions should be qualified. We conducted our study in students with an average age of 20.89 in Study 1 and 20.36 in Study 2. Age is known to affect people's thermoregulatory abilities, particularly in older age (e.g. [77]). We suspect that throughout different age groups—except the elderly—the relationship between social thermoregulation desires, attachment avoidance, and loneliness will hold. As in late adolescence [78], loneliness is a pressing topic. It is in fact so pressing that it has been made a priority by the World Health Organization [79]. It is all the more important that our study is replicated in older samples to understand whether our effect will again replicate.

Furthermore, our population lives in the French Alps in Grenoble, an area that is moderately cold in the winter (e.g. average low temperature −1.2°C and average high temperature 5.9°C in January) and warm in the summer (e.g. average low temperature 14.2°C and average high temperature of 26.9°C; [80]). We suspect that the effect size for the link between social thermoregulation desires, attachment avoidance, and loneliness is either the same or larger in colder climates and may be weaker in warmer climates. Relatedly, the STRAQ-1 instrument that we use may not be only limitedly applicable to other populations, as some of the subscales of the STRAQ-1 scored poorly on measurement equivalence across different populations that were tested [29]. The Dujols *et al.* [75] project we mention above also intends to tackle that problem as people from different populations generated the items for the new STRAEQ-2 instrument.

One other limitation of our study is that our measurement focused on between-person, nomothetic assessments of personality and no individual, idiographic assessments across time and contexts (e.g. [81]). It is very likely that people vary across the year and seasons in how much they desire social thermoregulation and to be close to other people and this should be a focus for future prospective studies. It is also worth clarifying that self-reported social thermoregulatory desires may differ from actual thermoregulatory behaviours to an extent that is still unknown. Further, we took a very crude approach to exploring the factor structures as (1) the instruments have rarely (if ever) been tested among French populations and (2) our approach was intended to be as exploratory as possible. Indeed, the factor structure of a trait can vary at both the inter- and intra-individual level [66]. Such changes in the factor structure are even more plausible as we did not make the distinction between items measuring traits (probably more stable, e.g. 'I worry about things') and items reflecting states (e.g. 'I was worried about situations in which I might panic and make a fool of myself'). A better understanding of such trait–state variations should be better addressed in the future, especially since mediational processes are time-dependent processes.

## 11.3. Statistical versus causal mediation

Related to the issue of time dependence is the distinction between statistical and causal mediation. Cross-sectional—and probably biased—mediation parameters can give rise to different combinations of longitudinal mediation parameters [44]. This makes it hard, if not impossible, to capture the true mediation parameters in addition to the causality. By using the joint-significance test, our analyses probably do a better job at estimating the effect sizes in comparison to other mediation tests [72]. However, there are still considerable limitations in relation to causal inference. Even though the conditional random forest models at least exclude some relevant variables of interest as potential explanatory variables, there are many other potential variables that we did not test. In other words, unobserved mediators could have biased estimates of mediation effects and alternative causal paths may in fact exist [82]. A next step in identifying causal relationships between social thermoregulation and loneliness can be done through prospective studies, for example, using a longitudinal mediation structural equation modelling approach [83]. We see such studies as a priority, as loneliness is a key variable in explaining people's health.

## 11.4. Social thermoregulation desires and loneliness: need for prospective studies

Peoples' health and longevity strongly relate to the extent to which they are socially connected, this relationship being stronger than, for instance, obesity and health or frequent alcohol usage and health, even when controlled for by many other relevant lifestyle variables (see, for instance, [84]). The US Surgeon General Vivek Murthy found loneliness so troubling that he dubbed it a strategic priority [85]. Despite that the UK government dedicated a minister to social contact [86] and despite near-immediate access to social media, loneliness has not decreased over time [87]. Could it be that we may not completely understand the sociobiological origins of loneliness?

Somewhat counterintuitively, the regulation of core body temperature has been linked with the reasons why people form and maintain relationships (see e.g. Rocha IJzerman [88]). The present studies provide very robust evidence for a link between social thermoregulation desires and loneliness: in three different datasets, we keep finding this link and we find that this link is mediated by a lack of trust in romantic partners (i.e. attachment avoidance). We thus qualify our evidence at Evidence Readiness Level 5: candidate solutions in observational settings have been compared and formal predictions for positive expected effects have been generated (although unintended consequences have not yet been observed; [73]).

We are thus sufficiently confident to recommend investing resources into prospective studies to move to Evidence Readiness Level 6 to establish causal inference in a laboratory environment or a hospital, investigating first (1) whether social thermoregulation desires early in life (e.g. from parent to infant) relate to attachment avoidance and feelings of loneliness when the child grows up, after which (2) temperature interventions can be tested to examine whether social thermoregulation desires can be manipulated in adulthood. Prospective studies can be conducted by, for instance, recording newborn infant cries [89,90], relating them to peripheral temperature of parents [91], after which attachment can be assessed between 9 and 18 months through the strange situation [92]. If such links can be established, temperature can be manipulated to simulate secure attachment through temperature actuators that are responsive to infant cries (e.g. [93]).

But for now, the link between social thermoregulation, attachment, and loneliness at the stage of a statistical 'proto-theory' (i.e. presumably robust statistical effects but without clear specification of mechanisms). Our conservative approach allowed us to draw strong claims about the replicability of a particular statistical pattern in a particular population. However, there is still the need for theory formalization based on valid measures and proper theory building approach to ensure clear and testable predictions [94]. If this endeavour is met, this can constitute a stepping stone toward further interventions crafting to act effectively on feelings of loneliness.

## 12. Conclusion

So far, this study is the first examining the link between thermoregulation strategies and personality. We tested different mediation models through a strict cross-validation and out-of-sample testing approach, and we found robust evidence for the link between social thermoregulation desires, attachment avoidance, and loneliness. Our study can be the starting point to understand the biological origins of loneliness and to create interventions for the health-promoting effect of social relationships. As a matter of public health, future studies should investigate this possibility.

Ethics. We will include our ethics statement in our revision. Participants provided informed consent, but, in line with guidelines in our faculty, we did not have to obtain IRB approval for this study.
Data accessibility. The data and scripts can be found at https://osf.io/74fr3/.
Authors' contributions. A.W.: conceptualization, data curation, formal analysis, investigation, methodology, project administration, software, writing—original draft, writing—review and editing; M.B.: data curation, formal analysis, software, writing—review and editing; O.D.: formal analysis, validation, writing—review and editing; P.F.: writing—review and editing; H.IJ.: conceptualization, data curation, formal analysis, funding acquisition, investigation, methodology, project administration, resources, software, supervision, validation, writing—original draft, writing—review and editing.

All authors gave final approval for publication and agreed to be held accountable for the work performed therein.
Conflict of interest declaration. Hans IJzerman has written a popular science book about social thermoregulation (https://wwnorton.com/books/heartwarming).
Funding. The preparation of this work was partly funded by French National Research Agency 'Investissements d'avenir' program (grant no. ANR-15-IDEX-02) awarded to H.IJ. The project page is available at https://osf.io/74fr3/.

## Appendix A: Excluded items from the created scales and their rationale for exclusion

| created variable | items | original item scale | factor loading | rationale for exclusion |
|---|---|---|---|---|
| Anxiety 'Modified' | I worry that I won't measure up to other people | ECR anxiety | a | Cross-loading with self-esteem |
| | My partner only seems to notice me when I'm angry | ECR anxiety | 0.17 | Insufficient loading |
| Avoidance 'Modified' | | | | No items excluded |
| Anxiousness | I found it hard to wind down | DASS | 0.34 | Cross-loading with Well-being |
| | I experienced breathing difficulty (e.g. excessively rapid breathing, breathlessness in the absence of physical exertion) | DASS | 0.35 | Cross-loading with Well-being |
| | I tended to over-react to situations | DASS | 0.33 | Cross-loading with Well-being |
| | I felt that I was using a lot of nervous energy | DASS | 0.43 | Cross-loading with Well-being |
| | I was intolerant of anything that kept me from getting on with what I was doing | DASS | 0.32 | Cross-loading with Well-being |
| | In the last month, how often have you felt nervous and 'stressed'? | Self-reported Stress | −0.32 | Cross-loading with Well-being |
| Well-being | I found it hard to wind down | DASS | −0.61 | Cross-loading with Anxiousness |
| | I experienced breathing difficulty (e.g. excessively rapid breathing, breathlessness in the absence of physical exertion) | DASS | −0.41 | Cross-loading with Anxiousness |

(Continued.)

**27**

| created variable | items | original item scale | factor loading | rationale for exclusion |
|---|---|---|---|---|
| | I tended to over-react to situations | DASS | −0.34 | Cross-loading with Anxiousness |
| | I felt that I was using a lot of nervous energy | DASS | −0.47 | Cross-loading with Anxiousness |
| | I was intolerant of anything that kept me from getting on with what I was doing | DASS | −0.32 | Cross-loading with Anxiousness |
| | In the last month, how often have you felt nervous and 'stressed'? | Self-reported Stress | 0.47 | Cross-loading with Anxiousness |
| Self-esteem | I've been feeling good about myself | Warwick-Edinburgh Mental Well-Being Scale | 0.31 | Cross-loading with Self-esteem |
| | I've been feeling confident | Warwick-Edinburgh Mental Well-Being Scale | 0.48 | Cross-loading with Self-esteem |
| | Love life | IPIP-NEO Extraversion | −0.33 | Does not ostensibly measure the main construct |
| | I worry that I won't measure up to other people | ECR anxiety | 0.41 | Cross-loading with Anxiety 'Modified' |
| | I compare myself with other people | Repetitive Thinking Mode Questionnaire | 0.32 | Cross-loading with Trust |
| | I've been feeling confident | Warwick-Edinburgh Mental Well-Being Scale | −0.36 | Cross-loading with Well-being |
| Self-discipline | I am able to work effectively toward long-term goals | Self-control | −0.31 | Cross-loading with Leadership |
| Loneliness 'Modified' | I am an outgoing person | UCLA Loneliness | 0.19 | Insufficient loading |
| | I am unhappy being so withdrawn | UCLA Loneliness | 0.28 | Insufficient loading |
| Social Dominance Orientation 'Modified' | Feel sympathy for those who are worse off than myself | IPIP-NEO Agreeableness | 0.34 | Cross-loading with Empathy |
| | To get ahead in life, it is sometimes necessary to step on other groups | Social Dominance Orientation | 0.28 | Insufficient loading |
| | Sometimes other groups must be kept in their place | Social Dominance Orientation | 0.24 | Insufficient loading |
| | If certain groups of people stayed in their place, we would have fewer problems | Social Dominance Orientation | 0.13 | Insufficient loading |
| | In getting what your group wants, it is sometimes necessary to use force against other groups. | Social Dominance Orientation | 0.21 | Insufficient loading |
| | Some groups of people are just more worthy than others | Social Dominance Orientation | 0.31 | Cross-loading with Empathy |
| | Everyone should have their own lifestyle, religious beliefs, and sexual preferences, even if it makes them different from everyone else | Right Wing Authoritarianism | −0.36 | Cross-loading with Right Wing Authoritarianism 'Modified' |
| Right Prejudice | | | | No items excluded |
| Impulsiveness | Am not interested in theoretical discussions | IPIP-NEO Openness | −0.36 | Does not ostensibly measure the main construct |
| Reflection | | | | No items excluded |
| Trust | Keep others at a distance | IPIP-NEO Extraversion | 0.34 | Cross-loading with Sociability |
| | I compare myself with other people | Repetitive Thinking Mode Questionnaire | 0.31 | Cross-loading with Self-esteem |
| Right Wing Authoritarianism 'Modified' | Everyone should have their own lifestyle, religious beliefs, and sexual preferences, even if it makes them different from everyone else | Right Wing Authoritarianism | 0.41 | Cross-loading with Social Dominance Orientation 'Modified' |
| | Some of the best people in our country are those who are challenging our government, criticizing religion, and ignoring the 'normal way things are supposed to be done' | | 0.18 | Insufficient loading |
| | There are many radical, immoral people in our country today, who are trying to ruin it for their own godless purposes, whom the authorities should put out of action | | 0.18 | Insufficient loading |
| | A 'woman's place' should be wherever she wants to be. The days when women are submissive to their husbands and social conventions belong strictly in the past | | 0.25 | Insufficient loading |
| | There is no 'one right way' to live life; everybody has to create their own way | | 0.05 | Insufficient loading |
| Stimulation | | | | No items excluded |
| Sociability | Keep others at a distance | IPIP-NEO Extraversion | 0.38 | Cross-loading with Trust |
| | Prefer variety to routine | IPIP-NEO Openness | 0.34 | Does not ostensibly measure the main construct |
| | Seek adventure | IPIP-NEO Extraversion | 0.33 | Does not ostensibly measure the main construct |
| Leadership | Use others for my own ends | IPIP-NEO Agreeableness | −0.30 | Cross-loading with Empathy |
| | Yell at people | IPIP-NEO Agreeableness | −0.30 | Cross-loading with Impulsiveness |

| created variable | items | original item scale | factor loading | rationale for exclusion |
|---|---|---|---|---|
| | I understand very quickly what's going on inside me | Repetitive Thinking Mode Questionnaire | 0.32 | Does not ostensibly measure the main construct |
| | I have trouble concentrating | Self-control | −0.30 | Does not ostensibly measure the main construct |
| Empathy | Feel sympathy for those who are worse off than myself | IPIP-NEO Agreeableness | 0.32 | Cross-loading with Social Dominance Orientation 'Modified' |

# Appendix B: Created attachment and personality variables with their reliabilities from the training set

| created variable | items | original item scale | factor loading | rationale for exclusion |
|---|---|---|---|---|
| Anxiety 'Modified' | I worry that I won't measure up to other people | ECR anxiety | 0.44 | Cross-loading with self-esteem |
| | My partner only seems to notice me when I'm angry | ECR anxiety | 0.17 | Insufficient loading |
| Avoidance 'Modified' | | | | No items excluded |
| Anxiousness | I found it hard to wind down | DASS | 0.34 | Cross-loading with Well-being |
| | I experienced breathing difficulty (e.g. excessively rapid breathing, breathlessness in the absence of physical exertion) | DASS | 0.35 | Cross-loading with Well-being |
| | I tended to over-react to situations | DASS | 0.33 | Cross-loading with Well-being |
| | I felt that I was using a lot of nervous energy | DASS | 0.43 | Cross-loading with Well-being |
| | I was intolerant of anything that kept me from getting on with what I was doing | DASS | 0.32 | Cross-loading with Well-being |
| | In the last month, how often have you felt nervous and 'stressed'? | Self-reported Stress | −0.32 | Cross-loading with Well-being |
| Well-being | I found it hard to wind down | DASS | −0.61 | Cross-loading with Anxiousness |
| | I experienced breathing difficulty (e.g. excessively rapid breathing, breathlessness in the absence of physical exertion) | DASS | −0.41 | Cross-loading with Anxiousness |
| | I tended to over-react to situations | DASS | −0.34 | Cross-loading with Anxiousness |
| | I felt that I was using a lot of nervous energy | DASS | −0.47 | Cross-loading with Anxiousness |
| | I was intolerant of anything that kept me from getting on with what I was doing | DASS | −0.32 | Cross-loading with Anxiousness |
| | In the last month, how often have you felt nervous and 'stressed'? | Self-reported Stress | 0.47 | Cross-loading with Anxiousness |
| Self-esteem | I've been feeling good about myself | Warwick-Edinburgh Mental Well-Being Scale | 0.31 | Cross-loading with Self-esteem |
| | I've been feeling confident | Warwick-Edinburgh Mental Well-Being Scale | 0.48 | Cross-loading with Self-esteem |
| | Love life | IPIP-NEO Extravertion | −0.33 | Does not ostensibly measure the main construct |
| | I worry that I won't measure up to other people | ECR anxiety | 0.41 | Cross-loading with Anxiety 'Modified' |
| | I compare myself with other people | Repetitive Thinking Mode Questionnaire | 0.32 | Cross-loading with Trust |
| | I've been feeling confident | Warwick-Edinburgh Mental Well-Being Scale | −0.36 | Cross-loading with Well-being |

(Continued.)

| created variable | items | original item scale | factor loading | rationale for exclusion |
|---|---|---|---|---|
| Self-discipline | I am able to work effectively toward long-term goals | Self-control | −0.31 | Cross-loading with Leadership |
| Loneliness 'Modified' | I am an outgoing person | UCLA Loneliness | 0.19 | Insufficient loading |
|  | I am unhappy being so withdrawn | UCLA Loneliness | 0.28 | Insufficient loading |
| Social Dominance Orientation 'Modified' | Feel sympathy for those who are worse off than myself | IPIP-NEO Agreeableness | 0.34 | Cross-loading with Empathy |
|  | To get ahead in life, it is sometimes necessary to step on other groups | Social Dominance Orientation | 0.28 | Insufficient loading |
|  | Sometimes other groups must be kept in their place | Social Dominance Orientation | 0.24 | Insufficient loading |
|  | If certain groups of people stayed in their place, we would have fewer problems | Social Dominance Orientation | 0.13 | Insufficient loading |
|  | In getting what your group wants, it is sometimes necessary to use force against other groups. | Social Dominance Orientation | 0.21 | Insufficient loading |
|  | Some groups of people are just more worthy than others | Social Dominance Orientation | 0.31 | Cross-loading with Empathy |
|  | Everyone should have their own lifestyle, religious beliefs, and sexual preferences, even if it makes them different from everyone else | Right Wing Authoritarianism | −0.36 | Cross-loading with Right Wing Authoritarianism 'Modified' |
| Right Prejudice |  |  |  | No items excluded |
| Impulsiveness | Am not interested in theoretical discussions | IPIP-NEO Openness | −0.36 | Does not ostensibly measure the main construct |
| Reflection |  |  |  | No items excluded |
| Trust | Keep others at a distance | IPIP-NEO Extraversion | 0.34 | Cross-loading with Sociability |
|  | I compare myself with other people | Repetitive Thinking Mode Questionnaire | 0.31 | Cross-loading with Self-esteem |
| Right Wing Authoritarianism 'Modified' | Everyone should have their own lifestyle, religious beliefs, and sexual preferences, even if it makes them different from everyone else | Right Wing Authoritarianism | 0.41 | Cross-loading with Social Dominance Orientation 'Modified' |
|  | Some of the best people in our country are those who are challenging our government, criticizing religion, and ignoring the 'normal way things are supposed to be done' |  | 0.18 | Insufficient loading |
|  | There are many radical, immoral people in our country today, who are trying to ruin it for their own godless purposes, whom the authorities should put out of action |  | 0.18 | Insufficient loading |
|  | A 'woman's place' should be wherever she wants to be. The days when women are submissive to their husbands and social conventions belong strictly in the past |  | 0.25 | Insufficient loading |
|  | There is no 'one right way' to live life; everybody has to create their own way |  | 0.05 | Insufficient loading |
| Stimulation |  |  |  | No items excluded |
| Sociability | Keep others at a distance | IPIP-NEO Extraversion | 0.38 | Cross-loading with Trust |
|  | Prefer variety to routine | IPIP-NEO Openness | 0.34 | Does not ostensibly measure the main construct |
|  | Seek adventure | IPIP-NEO Extraversion | 0.33 | Does not ostensibly measure the main construct |
| Leadership | Use others for my own ends | IPIP-NEO Agreeableness | −0.30 | Cross-loading with Empathy |
|  | Yell at people | IPIP-NEO Agreeableness | −0.30 | Cross-loading with Impulsiveness |
|  | I understand very quickly what's going on inside me | Repetitive Thinking Mode Questionnaire | 0.32 | Does not ostensibly measure the main construct |
|  | I have trouble concentrating | Self-control | −0.30 | Does not ostensibly measure the main construct |
| Empathy | Feel sympathy for those who are worse off than myself | IPIP-NEO Agreeableness | 0.32 | Cross-loading with Social Dominance Orientation 'Modified' |

# Appendix C. Differences in items part of the personality factors Anxiousness and Loneliness 'Modified' when comparing Study 1 and 2.

| Personality factor | Study 1 | | Study 2 | |
| --- | --- | --- | --- | --- |
| | Items | Original item scale | Missing items | Original item scale |
| Anxiousness | I experienced trembling (eg, in the hands) | Depression Anxiety Stress Scale | I experienced trembling (eg, in the hands) | Depression Anxiety Stress Scale |
| | I was worried about situations in which I might panic and make a fool of myself | Depression Anxiety Stress Scale | I was worried about situations in which I might panic and make a fool of myself | Depression Anxiety Stress Scale |
| | I felt I was close to panic | Depression Anxiety Stress Scale | I felt I was close to panic | Depression Anxiety Stress Scale |
| | I was aware of the action of my heart in the absence of physical exertion (eg, sense of heart rate increase, heart missing a beat | Depression Anxiety Stress Scale | I was aware of the action of my heart in the absence of physical exertion (eg, sense of heart rate increase, heart missing a beat | Depression Anxiety Stress Scale |
| | I felt scared without any good reason | Depression Anxiety Stress Scale | I felt scared without any good reason | Depression Anxiety Stress Scale |
| | Remain calm under pressure | IPIP-NEO Neuroticism | | |
| | Worry about things | IPIP-NEO Neuroticism | | |
| | Fear for the worst | IPIP-NEO Neuroticism | | |
| | Am afraid of many things | IPIP-NEO Neuroticism | | |
| | Get stressed out easily | IPIP-NEO Neuroticism | | |
| | Panic easily | IPIP-NEO Neuroticism | | |
| | Become overwhelmed by events | IPIP-NEO Neuroticism | | |
| | I feel pressured to keep my worst fears from coming | Repetitive Thinking Mode Questionnaire | | |
| Loneliness 'Modified' | I feel in tune with the people around me | UCLA Loneliness | I've been feeling close to other people | Warwick-Edinburgh Mental Well-Being Scale |
| | I lack companionship | UCLA Loneliness | | |
| | There is no one I can turn to | UCLA Loneliness | | |
| | I do not feel alone | UCLA Loneliness | | |
| | I feel part of a group of friends | UCLA Loneliness | | |
| | I have a lot in common with the people around me | UCLA Loneliness | | |
| | I am no longer close to anyone | UCLA Loneliness | | |
| | My interests and ideas are not shared by those around me | UCLA Loneliness | | |
| | There arc people I feel close to | UCLA Loneliness | | |
| | I feel left out | UCLA Loneliness | | |
| | My social relationships arc superficial | UCLA Loneliness | | |
| | No one really knows me well | UCLA Loneliness | | |
| | I feel isolated from others | UCLA Loneliness | | |
| | I can find companionship when I want it | UCLA Loneliness | | |
| | There are people who really understand me | UCLA Loneliness | | |
| | People are around me but not with me | UCLA Loneliness | | |
| | There are people I can talk to | UCLA Loneliness | | |
| | There are people I can turn to | UCLA Loneliness | | |
| | I've been feeling close to other people | Warwick-Edinburgh Mental Well-Being Scale | | |

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
