## [Peer Review File · Royal Society Open Science]

Review History

RSOS-201068.R0 (Original submission)

Review form: Reviewer 1 (Elizabeth Gross)

Do you have any ethical concerns with this paper?

No

Recommendation?

Accept with minor revision

Comments to the Author(s)

Please see the attached files for comments (see Appendix A).

Review form: Reviewer 2 (Lorne Campbell)

Do you have any ethical concerns with this paper?

No

Recommendation?

Accept with minor revision

Comments to the Author(s)

Thank you for the opportunity to review your research and research plans.

1. The scientific validity of the research question(s).

The approach taken to investigate the proposed research questions is, in my opinion, very sound. Moving from exploratory to confirmatory analytic tests, and including a new sample, allows for confidence to be placed in the final outcomes of the analyses. Great work.

2. The logic, rationale, and plausibility of the proposed hypotheses.

I admit that I am not an expert in the research on social thermoregulation. In my opinion, more rationale for the proposed mediation model would be useful. Also, do the authors think there could be regional differences given that some populations spent many, many years in relatively warm climates where others spent time in relatively cool/cold climates? Would the proposed model fit differently for people in, say, Central America compared to the French Alps?

3. The soundness and feasibility of the methodology and analysis pipeline (including statistical power analysis where applicable).

Overall I think the methodology and analysis pipeline overall is very sound and feasible. A true strength of this research.

4. Whether the clarity and degree of methodological detail would be sufficient to replicate the proposed experimental procedures and analysis pipeline.

5. Whether the authors provide a sufficiently clear and detailed description of the methods to prevent undisclosed flexibility in the experimental procedures or analysis pipeline.

In my opinion the authors are very clear and forthcoming, and provide links to their analyses and outcomes.

6. Whether the authors have considered sufficient outcome-neutral conditions (e.g. absence of floor or ceiling effects; positive controls; other quality checks) for ensuring that the results obtained are able to test the stated hypotheses.

It seems to me that they made these consideration.

I do have one concern that I think the authors should consider addressing. The research relies on a mediation model with the use of data collected at one point in time. There has been a lot written on the problems associated with these types of models, leaving some to wonder if there is much value to them at all. I am curious to read what the authors think of this debate and what it may mean for their proposed model with the data they have available.

Minor points

* In the first study presented all questions were answered online. Height and weight are therefore self-reported, not directly measured. It seems appropriate to label them as such.

* I think it would be valuable to include specific information regarding the power analysis for mediation so that it is reproducible. Exactly how was this analysis conducted?

* On page 12 I am guessing that most people will not be familiar with what is termed the "testweek". A few extra words here would add clarity.

* For the exploratory factor analysis, I am curious about the degree of item overlap across the many scales that were included.

* What approach was used to randomly split the data set into two data sets?

* On page 18 the authors say that the approach they used was more appropriate than linear regression because linear regression is parametric where their approach is non-parametric. Can they briefly elaborate *why* this is a good thing.

Decision letter (RSOS-201068.R0)

Dear Dr IJzerman

On behalf of the Editors, I am pleased to inform you that your Manuscript RSOS-201068 entitled "Personality as Adaption to Temperature (in French Students)" deemed suitable for in-principle acceptance in Royal Society Open Science subject to minor revision in accordance with the referee and editor suggestions. Please find their comments at the end of this email.

The reviewers and handling editors have recommended publication, but also suggest some minor revisions to your manuscript. Therefore, I invite you to respond to the comments and revise your manuscript.

Please you submit the revised version of your manuscript within 30 days (i.e. by the 16 September 2020). If you do not think you will be able to meet this date please let me know immediately.

Full author guidelines can be found here <https://royalsocietypublishing.org/rsos/registered-reports#ReviewerGuideRegRep>.

Kind regards
Professor Chris Chambers
Royal Society Open Science
openscience@royalsociety.org

on behalf of Professor Chris Chambers (Subject Editor, Royal Society Open Science)
openscience@royalsociety.org

Associate Editor Comments to Author (Professor Chris Chambers):

Associate Editor: 1

Comments to the Author:

Thank you for your patience during this challenging time for reviewers. Two reviewers have now assessed the manuscript. Both reviews are positive, while also requesting a range of revisions to

improve clarity of the rationale and methodology (including some structural amendments), and to justify the analytic approach taken (in particular, the use of mediation models). Provided the authors are able to respond comprehensively to all points raised, Stage 1 IPA should be forthcoming following either a rapid re-review or following assessment by the editors.

Reviewer comments to Author:

Reviewer: 1

Comments to the Author(s)

Please see the attached files for comments. (Review-RoyalSocietyOpenScience.docx)

Reviewer: 2

Comments to the Author(s)

Thank you for the opportunity to review your research and research plans.

1. The scientific validity of the research question(s).

The approach taken to investigate the proposed research questions is, in my opinion, very sound. Moving from exploratory to confirmatory analytic tests, and including a new sample, allows for confidence to be placed in the final outcomes of the analyses. Great work.

2. The logic, rationale, and plausibility of the proposed hypotheses.

I admit that I am not an expert in the research on social thermoregulation. In my opinion, more rationale for the proposed mediation model would be useful. Also, do the authors think there could be regional differences given that some populations spent many, many years in relatively warm climates where others spent time in relatively cool/cold climates? Would the proposed model fit differently for people in, say, Central America compared to the French Alps?

3. The soundness and feasibility of the methodology and analysis pipeline (including statistical power analysis where applicable).

Overall I think the methodology and analysis pipeline overall is very sound and feasible. A true strength of this research.

4. Whether the clarity and degree of methodological detail would be sufficient to replicate the proposed experimental procedures and analysis pipeline.

5. Whether the authors provide a sufficiently clear and detailed description of the methods to prevent undisclosed flexibility in the experimental procedures or analysis pipeline.

In my opinion the authors are very clear and forthcoming, and provide links to their analyses and outcomes.

6. Whether the authors have considered sufficient outcome-neutral conditions (e.g. absence of floor or ceiling effects; positive controls; other quality checks) for ensuring that the results obtained are able to test the stated hypotheses.

It seems to me that they made these consideration.

I do have one concern that I think the authors should consider addressing. The research relies on a mediation model with the use of data collected at one point in time. There has been a lot written on the problems associated with these types of models, leaving some to wonder if there is much value to them at all. I am curious to read what the authors think of this debate and what it may mean for their proposed model with the data they have available.

Minor points

* In the first study presented all questions were answered online. Height and weight are therefore self-reported, not directly measured. It seems appropriate to label them as such.

* I think it would be valuable to include specific information regarding the power analysis for mediation so that it is reproducible. Exactly how was this analysis conducted?

* On page 12 I am guessing that most people will not be familiar with what is termed the "testweek". A few extra words here would add clarity.

* For the exploratory factor analysis, I am curious about the degree of item overlap across the many scales that were included.

* What approach was used to randomly split the data set into two data sets?

* On page 18 the authors say that the approach they used was more appropriate than linear regression because linear regression is parametric where their approach is non-parametric. Can they briefly elaborate *why* this is a good thing.

Author's Response to Decision Letter for (RSOS-201068.R0)

See Appendix B.

Decision letter (RSOS-201068.R1)

Dear Dr IJzerman

On behalf of the Editor, I am pleased to inform you that your Manuscript RSOS-201068.R1 entitled "Personality as Adaption to Temperature (in French Students)" has been accepted in principle for publication in Royal Society Open Science.

You may now progress to Stage 2 and complete the study as approved. Before commencing data collection we ask that you:

- 1) Update the journal office as to the anticipated completion date of your study.
- 2) Register your approved protocol on the Open Science Framework (<https://osf.io/rr>) or other recognised repository, either publicly or privately under embargo until submission of the Stage 2 manuscript. Please note that a time-stamped, independent registration of the protocol is mandatory under journal policy, and manuscripts that do not conform to this requirement cannot be considered at Stage 2. The protocol should be registered unchanged from its current approved state, with the time-stamp preceding implementation of the approved study design.

Following completion of your study, we invite you to resubmit your paper for peer review as a Stage 2 Registered Report. Please note that your manuscript can still be rejected for publication at Stage 2 if the Editors consider any of the following conditions to be met:

- The results were unable to test the authors' proposed hypotheses by failing to meet the approved outcome-neutral criteria.
- The authors altered the Introduction, rationale, or hypotheses, as approved in the Stage 1 submission.
- The authors failed to adhere closely to the registered experimental procedures. Please note that any deviations from the approved experimental procedures must be communicated to the editor immediately for approval, and prior to the completion of data collection. Failure to do so can result in revocation of in-principle acceptance and rejection at Stage 2 (see complete guidelines for further information).

- Any post-hoc (unregistered) analyses were either unjustified, insufficiently caveated, or overly dominant in shaping the authors' conclusions.
- The authors' conclusions were not justified given the data obtained.

We encourage you to read the complete guidelines for authors concerning Stage 2 submissions at <https://royalsocietypublishing.org/rsos/registered-reports#ReviewerGuideRegRep>. Please especially note the requirements for data sharing, reporting the URL of the independently registered protocol, and that withdrawing your manuscript will result in publication of a Withdrawn Registration.

Please note that Royal Society Open Science will introduce article processing charges for all new submissions received from 1 January 2018. Registered Reports submitted and accepted after this date will ONLY be subject to a charge if they subsequently progress to and are accepted as Stage 2 Registered Reports. If your manuscript is submitted and accepted for publication after 1 January 2018 (i.e. as a full Stage 2 Registered Report), you will be asked to pay the article processing charge, unless you request a waiver and this is approved by Royal Society Publishing. You can find out more about the charges at <https://royalsocietypublishing.org/rsos/charges>. Should you have any queries, please contact openscience@royalsociety.org.

Once again, thank you for submitting your manuscript to Royal Society Open Science and we look forward to receiving your Stage 2 submission. If you have any questions at all, please do not hesitate to get in touch. We look forward to hearing from you shortly with the anticipated submission date for your stage two manuscript.

on behalf of Professor Chris Chambers (Registered Reports Editor, Royal Society Open Science)
openscience@royalsociety.org

Author's Response to Decision Letter for (RSOS-201068.R1)

See Appendix C.

RSOS-201068.R2

Review form: Reviewer 1 (Elizabeth Gross)

Is the manuscript scientifically sound in its present form?

Yes

Are the interpretations and conclusions justified by the results?

Yes

Is the language acceptable?

Yes

Do you have any ethical concerns with this paper?

No

Have you any concerns about statistical analyses in this paper?

No

Recommendation?

Accept with minor revision

Comments to the Author(s)

Please see the attached file for my comments to the authors (see Appendix D).

Decision letter (RSOS-201068.R2)

Dear Dr IJzerman:

It is a pleasure to accept your Stage 2 Registered Report entitled "Individual differences in adapting to temperature in French students are only related to attachment avoidance and loneliness" in its current form for publication in Royal Society Open Science. The comments of the reviewer(s) who reviewed your manuscript are included at the foot of this letter.

Thank you for your fine contribution. On behalf of the Editors of Royal Society Open Science, we look forward to your continued contributions to the journal.

on behalf of Professor Chris Chambers (Associate Editor) and Chris Chambers (Subject Editor)
openscience@royalsociety.org

Associate Editor Comments to Author (Professor Chris Chambers):
Comments to the Author:

One of the original reviewers from Stage 1 was available to evaluate the Stage 2 manuscript within a reasonable time frame, and I have decided that we can proceed with a decision based on this reviewer's assessment and my own reading of the manuscript. As you will see, the reviewer is very positive about the submission, noting the careful and rigorous implementation of the approved protocol, and that the conclusions are appropriately aligned to the evidence. I fully agree - this is an impressive RR that clearly meets the Stage 2 criteria. The reviewer offers some minor comments for revision, but in light anything more substantive, I don't think it is necessary to introduce a revision round. I have therefore decided to issue an Accept decision and recommend the authors consider these minor corrections at the proof stage. Congratulations on an outstanding piece of work.

Reviewer comments to Author:

Reviewer: 1

Comments to the Author(s)

Please see the attached file for my comments to the authors
(**Review_RoyalSocietyOpenScience_Phase2.pdf**).

Appendix A

Review of "Personality as Adaption to Temperature (in French Students)"

July 6th, 2020

The authors aim to establish whether individual differences in personality are predicted by social and solitary thermoregulation, with attachment style as a potential mediator. The proposed study is novel and interesting, extending the existing work on social thermoregulation in a new direction, and impacts other important fields in psychology – personality and attachment theory. Their hypothesis, while admittedly exploratory, has support from both comparative studies in animal behavior and biology, and behavioral studies in thermoregulation in humans. Except for a few minor clarifications (see below), I found the introduction easy to follow and their argument well-reasoned. The methodology, experimental procedure, and proposed analysis pipeline was appropriately detailed for direct replication. The proposed analysis plan is very detailed, with explicit guidelines for claiming a weak to strong replication. So long as the authors commit to either keeping the supplementary materials (analysis scripts, etc.) hosted on the OSF Framework indefinitely or providing copies to the journal for listing supplementary materials, I see no room for flexibility in analyses. As a sufficiently large sample has already been collected for confirmatory analyses, I also anticipate no issues in their ability to test their proposed hypotheses.

However, there are a few points of clarification and minor revisions the authors should address, outline below.

Points of Clarification

1. When discussing the predictive and reactive control strategy (beginning last paragraph on page 7 of the .pdf proof), the authors summarize a previous review article that states that individuals engage more in predictive control behaviors if raised in predictable environments, and vice versa. What exactly constitutes a predictable environment? I am inclined to think of factors such as SES, but upon reading (admittedly quickly) the cited article, this does not seem to be correct.
2. When introducing the STRAQ-1 (page 9 of the proof), the authors summarize previous research relating the desire for social thermoregulation to attachment avoidance. Is this to say that the STRAQ-1 did not previously relate to attachment anxiety? I would like to hear more about why it might not have and the rationale for including attachment anxiety then in the exploratory analyses.
3. Could you clarify what constitutes a 'comfortable climate' in summarizing Wei and colleagues' work (page 10 of the proof)?
4. There is a large discrepancy in gender breakdown of the sample. For an admittedly exploratory study, it was surprising that there was no discussion of possible gender differences. Could the authors comment on why they chose to leave out this particular analysis? For example, do they not expect gender to interact with any of the variables, or given the samples have already been

collected, did they feel that the low number of males prohibited statistical analysis?

5. Likewise, the STRAQ-1 items seem as if they might be correlated with relationship status, and this is purely speculative, but I am also wondering if other personality and attachment variables would relate to relationship status. Were any analyses run to investigate this relationship? If not, is there a rationale to not do so?
6. The paragraph following Figure 2, describing conditional random forest analysis, was very helpful in clarifying the procedure. It should be moved to earlier in the paper when the process is first introduced.
7. In Table 3, you list that social dominance orientation 'modified', reflection, and self-discipline did not have any listed predictors or mediators, yet you list a planned analysis as 'correlation'. Correlation with what variables, and why? I gather that social dominance 'modified' was predicted by solitary thermoregulation in your exploratory analysis of mediation pathway c, but I did not see where reflection and self-discipline had any results in the exploratory analyses.
8. Tables 4 and 5 are hard to read, I would suggest lines separating the rows of the tables.

Minor typos and misspellings

1. Page 11 line 181 states "This meant that we first a large...". I believe you are missing a verb (identified?).
2. Page 11 line 193 "a priority" predictions should be "a priori".
3. Page 12, line 212 "As per the guidelines for Registered Reports, we do not yet analyze..." is awkward phrasing. Perhaps consider "...we have not yet analyzed" as a replacement.
4. You are missing a period at the end of the sentence on line 280, page 17.
5. The word 'mediation' is misspelled on page 21, line 363.
6. On page 27, line 468, you list the results of the Monte Carlo method in Table 3, when it is reported in Table 4.

Appendix B

Dear Dr. Chambers,

Thank you for your editorial letter dated August 17th 2020 regarding our manuscript Personality as Adaptation to Temperature (in French Students). In this response letter, we detail how we have dealt with your and the reviewers' comments. We have also created a version with track changes accepted and track changes on so that you can more easily see what we have changed.

Reviewer 1: When discussing the predictive and reactive control strategy (beginning last paragraph on page 7 of the .pdf proof), the authors summarize a previous review article that states that individuals engage more in predictive control behaviors if raised in predictable environments, and vice versa. What exactly constitutes a predictable environment? I am inclined to think of factors such as SES, but upon reading (admittedly quickly) the cited article, this does not seem to be correct.

Authors' Response: PARCS in general refers to predictable environments that are socially predictable (e.g., when people when young experience a secure environment from the caregiver; see e.g., Tops et al., 2014). What constitutes predictable environments has not been well explored yet. We are interpreting predictability of environments based on work by Vergara et al. (2018), who assumes that environments that present greater physical dangers (e.g., threat of bodily harm) or “climatic dangers” (harsh climates) can represent unpredictable environments. SES could certainly be a part of this, at least from our perspective. This is largely untested (and is on our agenda to test in the future). To clarify this, we have adjusted this part in our text (the highlighted portion changed):

“Clusters of individual differences similar to animal temperaments also manifest in humans. According to Predictive And Reactive Control Systems (PARCS) theory, people differ in their

tendency to rely on predictive versus reactive control – two systems that resemble proactive and reactive animal temperaments depending on the predictability of the environment (Tops, Boksem, Quirin, & Koole, 2014; Tops et al., 2020). An environment is thought to be predictable when it is not threatening or when threats are manageable (possibly including, for example, bodily harm, harsh climates, or limited economic means). People engage more in predictive control when processing familiar stimuli and tend to engage more if they are raised in predictable environments. People engage more in reactive control when processing novel information and if they are raised in more unpredictable environments. Predictive control is considered more metabolically efficient than reactive control because it allows people to schedule more in advance (Tops et al., 2020).”

Further, please also note that in our previous draft we had written something that relates to this point: “Also, we don’t know exactly how to measure “life-history strategies” in humans, thus supplying the need to step back and explore when testing our general ideas in a French sample.”

Reviewer 1: When introducing the STRAQ-1 (page 9 of the proof), the authors summarize previous research relating the desire for social thermoregulation to attachment avoidance. Is this to say that the STRAQ-1 did not previously relate to attachment anxiety? I would like to hear more about why it might not have and the rationale for including attachment anxiety then in the exploratory analyses.

Authors’ Response: In Vergara et al. (2019) solitary thermoregulation was not significantly related to attachment avoidance but was significantly related to attachment anxiety with $r = .08, p < .001$. We thus decided to include solitary thermoregulation in our exploratory analyses. Another rationale to include attachment anxiety is that our sample was an out-of-sample test (in a population not yet tested before, French students). As a consequence, we felt

appropriate to explore how attachment anxiety related to our STRAQ-1 variables in our sample. To clarify this, we added the link between solitary thermoregulation and attachment in the text (the highlighted portion changed).

“The STRAQ-1 has for example a subscale that reliably assesses individual differences in terms of desires to socially thermoregulate. The STRAQ-1 was created and validated to investigate whether people’s strategies to thermoregulate relate to their feelings of reliability and safety in relationships. And indeed, Vergara et al. (2019) found that people’s desires to socially and solitarily thermoregulate respectively relate to avoidance and anxiety in relationships. The STRAQ-1 thus provides not only a valid measurement tool, but also shows that people’s ways of coping with the environment relate to their attachment styles. This is thus a first indication that the way people cope with the environment could potentially shape their personality.”

Reviewer 1: Could you clarify what constitutes a ‘comfortable climate’ in summarizing Wei and colleagues’ work (page 10 of the proof)?

Authors’ response: Wei and colleagues defined a *clement* (not comfortable) climate as being close to 22°C. We have now adjusted this in the text.

“Wei and colleagues found that people are more agreeable, extraverted, conscientious, open to experience, and less neurotic if they grew up in clement climates (closer to the psychophysiological comfort optimum of 22°C)”

Reviewer 1: There is a large discrepancy in gender breakdown of the sample. For an admittedly exploratory study, it was surprising that there was no discussion of possible gender differences. Could the authors comment on why they chose to leave out this particular analysis? For example, do they not expect gender to interact with any of the variables, or

given the samples have already been collected, did they feel that the low number of males prohibited statistical analysis?

Authors' response: We understand the reviewer's surprise. However, if one wants to make an inference of just French males versus females, one needs a representative sample of that population or various samples (collected in a different way). For that reason, we did not include gender in the analyses. However, the reviewer is right that it should be a concern, as IJzerman et al. (2020) find a relationship between gender and effect size in their meta-analysis. They make the same recommendation (i.e., not to compare gender with non-representative samples), but to include gender as a control variable. For that reason, we will change to analyses with gender as control in the main text and without gender as control variable online for our exploratory and confirmatory datasets. We have now adjusted this in the following paragraph:

“For all mediation models, we decided to include the sex as control variable as per the recommendation by IJzerman et al. (2020). They found that participants' sex may have an influence on the effect size of variables related to social thermoregulation (we did not make any further inferences about sex, because our sample was not representative of men versus women). We include the analyses without sex as control variable on our OSF page (<https://osf.io/f6qun/>).”

Reviewer 1: Likewise, the STRAQ-1 items seem as if they might be correlated with relationship status, and this is purely speculative, but I am also wondering if other personality and attachment variables would relate to relationship status. Were any analyses run to investigate this relationship? If not, is there a rationale to not do so?

Authors' response: The reviewer is correct, as Vergara et al. 2019 found a correlation between relationship status and social thermoregulation desires. Although again out-of-

sample, we can expect a link between relationship status and social thermoregulation desires in our sample as well. Because they are somewhat tangential to our paper, we will report the correlation between relationship status and STRAQ-1 variables in our footnotes.

Reviewer 1: The paragraph following Figure 2, describing conditional random forest analysis, was very helpful in clarifying the procedure. It should be moved to earlier in the paper when the process is first introduced.

Authors' response: We inserted this paragraph earlier in the text, in the “Analysis strategy overview”. See below the paragraph in which we inserted conditional random forest description:

“Once we defined our new factors, we explored existing relations in our data to generate mediation model hypotheses through a powerful supervised machine learning method called conditional random forests. In supervised machine learning more generally, the algorithm infers a pattern from the data derived from a “signal” (or dependent variable). The method relies on “out-of-bag estimates”, which involves repeated sampling from a training dataset (e.g., Breiman, 2001). Multiple “trees” are formed by assessing whether each variable influences the “signal”. The “trees” (votes on whether variables matter for the outcome variable or not) are then assembled into a “forest”. Each “tree” receives a “vote” into an ensemble model that then summarizes all information from the trees. The outcome in the case of conditional random forests is a variable importance list. The importance list allows us to identify which are the best predictors of the variable of interest and which of the computed variables differ from random noise when predicting the variable of interest (see also IJzerman et al., 2016a). In our case the random forest allows us to select the variables to be included in the mediation analyses in the exploratory phase. In the first stage of our exploratory analyses,

we ran three sets of conditional random forests, one for each of the three mediation paths (the *a*, *b*, and *c* path).”

Reviewer 1: In Table 3, you list that social dominance orientation ‘modified’, reflection, and self-discipline did not have any listed predictors or mediators, yet you list a planned analysis as ‘correlation’. Correlation with what variables, and why? I gather that social dominance ‘modified’ was predicted by solitary thermoregulation in your exploratory analysis of mediation pathway *c*, but I did not see where reflection and self-discipline had any results in the exploratory analyses.

Authors’ response: This was a mistake on our part and we thank the reviewer for pointing this out. The variables reflection, self-discipline, and stimulation aren’t predicted by any variables. We have now replaced “Correlation” by “None” for these variables in the “resulting model” column. Social dominance orientation is indeed predicted by solitary thermoregulation so this was also a mistake on our part. We have now added solitary thermoregulation as a predictor of social dominance orientation as a predictor in the Table 3.

Reviewer 1: Tables 4 and 5 are hard to read, I would suggest lines separating the rows of the tables.

Authors’ response: As the reviewer suggested, we inserted lines to separate the rows of the table.

Reviewer 1: Minor typos and misspellings

1. Page 11 line 181 states “This meant that we first a large...”. I believe you are missing a verb (identified?).
2. Page 11 line 193 “a priority” predictions should be “a priori”.

3. Page 12, line 212 “As per the guidelines for Registered Reports, we do not yet analyze...” is awkward phrasing. Perhaps consider “...we have not yet analyzed” as a replacement.
4. You are missing a period at the end of the sentence on line 280, page 17.
5. The word ‘mediation’ is misspelled on page 21, line 363.
6. On page 27, line 468, you list the results of the Monte Carlo method in Table 3, when it is reported in Table 4.

Authors’ response: All these mistakes have been fixed. Thank you for helping us find these.

Reviewer: 2

Comments to the Author(s)

Thank you for the opportunity to review your research and research plans.

1. The scientific validity of the research question(s).

The approach taken to investigate the proposed research questions is, in my opinion, very sound. Moving from exploratory to confirmatory analytic tests, and including a new sample, allows for confidence to be placed in the final outcomes of the analyses. Great work.

2. The logic, rationale, and plausibility of the proposed hypotheses.

I admit that I am not an expert in the research on social thermoregulation. In my opinion, more rationale for the proposed mediation model would be useful. Also, do the authors think there could be regional differences given that some populations spent many, many years in relatively warm climates where others spent time in relatively cool/cold climates? Would the

proposed model fit differently for people in, say, Central America compared to the French Alps?

Authors' response: The current reasoning behind the proposed relationships is indeed speculative and restricted to the relative predictability of the environment (e.g., the temperature). But our data cannot speak to that idea in more detail because we neither have samples in different locations, nor did we do a longitudinal study. For that reason, we try to remain modest in our theoretical background and provide a proof-of-concept for *relationships* between different variables. The mechanisms thus remain speculative. What we were already planning to do is to insert a Constraints of Generality section in our discussion to highlight the limitations, where we would discuss exactly those issues (e.g., would the proposed model fit differently for peoples from different areas of the world). We agree that this relation could well depend on the region. Wei et al., for example, found that personality is influenced by climate. However, how climate exactly influences the mechanisms that we assumed needs to be further investigated. For now, we have not adjusted anything in our introduction related to this, but we do commit to discussing this in our Constraints on Generality section.

3. The soundness and feasibility of the methodology and analysis pipeline (including statistical power analysis where applicable).

Overall I think the methodology and analysis pipeline overall is very sound and feasible. A true strength of this research.

4. Whether the clarity and degree of methodological detail would be sufficient to replicate the proposed experimental procedures and analysis pipeline.

5. Whether the authors provide a sufficiently clear and detailed description of the methods to prevent undisclosed flexibility in the experimental procedures or analysis pipeline.

In my opinion the authors are very clear and forthcoming, and provide links to their analyses and outcomes.

6. Whether the authors have considered sufficient outcome-neutral conditions (e.g. absence of floor or ceiling effects; positive controls; other quality checks) for ensuring that the results obtained are able to test the stated hypotheses.

It seems to me that they made these considerations.

I do have one concern that I think the authors should consider addressing. The research relies on a mediation model with the use of data collected at one point in time. There has been a lot written on the problems associated with these types of models, leaving some to wonder if there is much value to them at all. I am curious to read what the authors think of this debate and what it may mean for their proposed model with the data they have available.

Authors' response: We thank the reviewer for raising this concern, as it is an important one. We briefly mentioned this issue in our previous version, for example on Page 9: “Although these mediation models cannot be taken as evidence of causal mechanisms given our cross-sectional design, they can serve as predictions in future studies with designs that support stronger causal inference”. However, we agree that there may be potential statistical issues beyond potential issues related to causal inference and we recognized that we could have done more to emphasize this problem. Our cross-sectional design may fail to capture the true parameters of the mediational processes. Notably, mediation with cross-sectional data can provide accurate parameter estimates *only* under certain conditions that are rarely met and that depend on assumptions about parameter changes over time (Maxwell & Cole, 2007; Maxwell, Cole, & Mitchell & 2011).

In our case, however, we attempt to shed light on causal mechanisms that may exist but do not draw any hasty conclusions on their existence. Indeed, we provide point predictions for the second half of the dataset. If our mediations are replicated, this can be a first indication that causal relations exist and this may help us to make predictions for studies with designs that are better able to provide causal inferences (such as experimental designs or longitudinal studies in which mediation processes are captured through longitudinal structural equation modelling methods; for a review see O'Laughlin, Martin, & Ferrer, 2018). We have added clarifications about the issues regarding cross-sectional design for mediational processes (see the highlighted portion below):

“These mediation models cannot be taken as evidence of causal mechanisms given that our data are cross-sectional and that mediational processes we assumed imply changes over time. For example, Maxwell and Cole (2007, 2011) demonstrated mathematically that the conditions under which cross-sectional designs may capture the true parameters of the mediation are rarely (if not never) met. Yet, despite the fact that our mediation does not reflect the causal processes we predict, they can indicate that some relations exist, particularly if replicated with the precision we predicted. Then still, a finding of statistical mediation in cross-sectional data does not imply the presence of causal relationships unless the analyst is willing to make strong, and unlikely unsupported (Maxwell & Cole, 2007; 2011), causal assumptions. This project thus aims to identify mediated relationships that *could* exist rather than relationships that *do* exist. The relationships that we identify can then be investigated in longitudinal and experimental studies that support stronger causal inference, in which some predictions may pan out and others will fail.”

*Minor points

* In the first study presented all questions were answered online. Height and weight are therefore self-reported, not directly measured. It seems appropriate to label them as such.

Authors' response: As suggested, we replaced “height” and “weight” by “self-reported height” and “self-reported weight”.

* I think it would be valuable to include specific information regarding the power analysis for mediation so that it is reproducible. Exactly how was this analysis conducted?

Authors' response: We added clarifications in the power analysis section. As well, we added the script of the power analysis on our OSF page and we added the link in the manuscript (see below the highlighted portions):

“To ensure we had sufficient power to run mediations in either half of our sample, before we conducted our analyses, we ran an a priori power analysis for mediation based on the Sobel test with the *powerMediation* package (Qiu, 2017). It first requires to estimate the standard deviation of the independent variable and of the mediator. It also requires to estimate the effect size of the relations between the independent variable and the relations between the mediator and the dependent variable. Finally, it requires to compute the standard deviation of the error term of the relation between the independent variable and the dependent variable controlling for the mediator based on the effect size of the three paths of the mediation (for more information, see Perugini, Gallucci, & Costantini, 2018). Based on a minimum effect size of interest of $\beta = .25$ for all relations in the mediation and a statistical power of .80, we found that 231 participants were needed to detect a mediation. We thus concluded that our number of participants was sufficient to be able to split our data in two equal parts (for our script and other details, see <https://osf.io/74fr3/>).”

* On page 12 I am guessing that most people will not be familiar with what is termed the "testweek". A few extra words here would add clarity.

Authors' response: We agree that this term may be misunderstood. We described more precisely what the "testweek" are in the text (see below the highlighted portion):

"Our measures were chosen by researchers who participated in the so-called "testweek" at Université Grenoble Alpes. Department members nominated questionnaires for inclusion (as these were thus not chosen by us, it constrained what we could find in our sample). The department then posted Qualtrics links with the full battery of questionnaires to student Facebook groups. Students could then participate in a battery of questions in exchange for course credits. Before moving onto our Exploratory Factor Analyses, we report the scales in their usual form together with their reliability from the training set (see Table 1; these will be updated to include the test set after review)."

* For the exploratory factor analysis, I am curious about the degree of item overlap across the many scales that were included.

Authors' response: The entire exploratory analysis is available at <https://osf.io/4qrvu/> with the items, the scale to which the items belong, and their load onto each factor. The factor loadings higher than .30 are highlighted in red (when opening it on excel), which we hope makes it easier to observe items' overlap in factor loading. We did not add anything in our manuscript, but we hope this answers the reviewer's question.

* What approach was used to randomly split the data set into two data sets?

Authors' response: We first randomized the data using an r script with a specific seed randomly chosen and then splitted it in half equals. We added the script that we used to split

the data at <https://osf.io/uehj9/> and specified in the text that the script was available on our OSF page.

* On page 18 the authors say that the approach they used was more appropriate than linear regression because linear regression is parametric where their approach is non-parametric. Can they briefly elaborate *why* this is a good thing.

Authors' response: We think that non-parametric approaches have various advantages over parametric approaches and especially in exploratory designs. In machine learning, a parametric algorithm relies on a fixed number of parameters. A non-parametric algorithm uses a flexible number of parameters and the number of parameters often grows as it learns more data. While slower, it makes fewer assumptions about the data, which is useful in highly exploratory designs. In our case, in which we had to explore many relations without any a priori knowledge about these relations or their linearities, non-parametric approaches are thus well-suited. We added clarifications in the text (see below the highlighted part):

“We chose conditional random forests over a regression for three reasons. First, linear regression is a parametric approach that requires a priori predictions of relationships between variables and also requires hypotheses regarding potential nonlinearities and interactions (Grömping, 2012). As we were interested in exploring our data, we relied on a non-parametric type of machine learning, which relies on a flexible number of parameters, where the number of parameters can grow as the algorithm learns more data (Russel & Norvig, 2002). Overall, they are therefore well-suited for situations in which researchers have no a priori predictions such as exploratory research. Second, random forests are less prone to overfitting in relatively small samples with multiple variables (Grömping, 2012). Third and finally, random forests have a smaller chance for collinearity when including multiple predictors, as is the case in the current situation (Matsuki, Kuperman, & Van Dyke, 2016).”

Additional change : We removed the interpretation of the mediation analyses (i.e. partial and full mediation) as these terms are arbitrary and may lead to erroneous conclusions about involved mediational processes (for a discussion, see Rucker, Preacher, Tormala, & Petty, 2011).

We hope to have addressed all the necessary concerns. We thank you for your review and hope that it now meets the high standards for Stage 1 Acceptance in *Royal Society Open Science*.

We remain available for any requests or questions.

Kind regards, also on behalf of the other co-authors,

Adrien Wittmann

References

- Maxwell, S. E., & Cole, D. A. (2007). Bias in cross-sectional analyses of longitudinal mediation. *Psychological Methods, 12*(1), 23–44. doi:10.1037/1082-989x.12.1.23
- Maxwell, S. E., Cole, D. A., & Mitchell, M. A. (2011) Bias in cross-sectional analyses of longitudinal mediation: Partial and complete mediation under an autoregressive model, multivariate. *Behavioral Research, 46*(5), 816-841, DOI: [10.1080/00273171.2011.606716](https://doi.org/10.1080/00273171.2011.606716)
- Kristine D. O'Laughlin, Monica J. Martin & Emilio Ferrer (2018): Cross-sectional analysis of longitudinal mediation processes. *Multivariate Behavioral Research*, DOI: [10.1080/00273171.2018.1454822](https://doi.org/10.1080/00273171.2018.1454822)
- Rucker, D. D., Preacher, K. J., Tormala, Z. L., & Petty, R. E. (2011). Mediation analysis in social psychology: Current practices and new recommendations. *Social and Personality Psychology Compass, 5*(6), 359–371. <https://doi.org/10.1111/j.1751-9004.2011.00355.x>
- Tops, M., Boksem, M. A., Quirin, M., IJzerman, H., & Koole, S. L. (2014). Internally directed cognition and mindfulness: An integrative perspective derived from predictive and reactive control systems theory. *Frontiers in Psychology, 5*, 429. <https://doi.org/10.3389/fpsyg.2014.00429>

Appendix C

Prof. Chris Chambers
Editor *Royal Society Open Science*

Menthon-Saint-Bernard, February 7, 2022

Objet : Wittmann et al.

Dear Prof. Chambers, dear Chris,

Attached you will find our resubmission of our *In Principle Accepted* report, originally entitled *Personality as Adaptation to Temperature (in French Students)*. Because of the eventual findings, we have changed our title to *Individual differences in adapting to temperature in French students are only related to attachment avoidance and loneliness*.

Following our IPA, we had pre-registered the project on the OSF at <https://osf.io/uqhsg/>. We then analyzed the dataset in our testing dataset. Once we had completed this step, we reported our analyses and updated our predictions in the following component: <https://osf.io/zx6pb/>. The second pre-registration (for the out-of-sample Study 2) can be found here: <https://osf.io/f8zgs>.

Thank you in advance for your consideration.

Cordially and on behalf of the co-authors,

Hans Rocha IJzerman, Ph. D.
Associate Professor of Psychology
Chair of the LIP/PC2S Social Cognition Group
Université Grenoble Alpes

Appendix D

Review of "Individual differences in adapting to temperature in French students are only related to attachment avoidance and loneliness"

March 22nd, 2022

The authors have conducted an exploratory study to investigate potential relationships between thermoregulation and personality variables. Using a large, cross-sectional dataset, they first generated mediation models for variables most likely to be related to thermoregulation, then attempted to replicate the models in two further datasets.

The Introduction, rationale, and stated hypothesis did not change from Stage 1 to Stage 2, and the data that was collected was sufficient to test hypothesis, based on a priori power analyses. The authors also adhered to the registered experimental procedures.

There was one unregistered, exploratory statistical analysis that was presented. Only one hypothesized model consistently replicated throughout the datasets; there was a relationship between Social Thermoregulation on Loneliness "Modified" mediated by Attachment Avoidance "Modified". As such, the authors wondered if the failure to replicate all models was due to a lack of power, and so combined both study datasets (N = 844) to test their proposed models. I found the rationale for this exploratory analysis to be sound. Furthermore, the authors were quite clear in the purpose of the exploratory analysis - hypothesis generation rather than confirmation. Therefore, I judge this exploratory analysis to be informative.

The authors' concluded that social thermoregulation predicted loneliness, mediated by attachment avoidance. Given their rigorous process of hypothesis-generation and then confirmation, I find this conclusion justified. I also appreciated the frank acknowledgment that the results are exploratory and the effect sizes are small. On the other hand, I am persuaded by the possibility that the social thermoregulation self-reports measures can be improved, and that age and sample's climate may also have created variability in the samples. As such, I feel their conclusions are well-reasoned and appropriate for the data.

However, there are a few minor revisions the authors should address, outlined below.

Minor typos and misspellings

1. Presumably, the original paper was written according to the 6th edition of the APA style manual. The authors should consider updating to the 7th edition (for example, changes in the running head, and changes in in-text citations for three or more authors).

2. Page 3 of the Introduction, line 57: "We then test the replicability" should be changed to reflect the past tense of the rest of the section, "We then tested the replicability."
3. Page 10 of the Introduction, line 214: "which" is misspelled with two h's.
4. In Table 1: under the Subscales for the STRAQ-1, "Social" is misspelled as "Socialt".
5. Page 37, line 650 and page 41, line 696: "May find these formalized predictions at..." is not a complete sentence and needs a subject.
6. Page 42, line 733: "the degree to which people know their confidence that others will be available" is oddly phrased, please consider rewriting it.